



# Ongoing firn warming at Eclipse Icefield, Yukon, indicates potential widespread meltwater percolation and retention in firn pack across the St. Elias Range

Ingalise Kindstedt[1,2], Dominic Winski[1,2], C. Max Stevens[3], Emma Skelton[1,2,4], Luke Copland[5], Karl Kreutz[1,2], Mikaila Mannello[1,2], Renée Clavette[1,2], Jacob Holmes[1,2], Mary Albert[6], and Scott N. Williamson[7]

[1]Climate Change Institute, University of Maine, Orono, Maine, USA
[2]School of Earth and Climate Sciences, University of Maine, Orono, Maine, USA
[3]Department of Earth and Space Sciences, University of Washington, Seattle, Washington, USA
[4]Cold Regions Research and Engineering Laboratory, Hanover, NH USA
[5]Department of Geography, Environment and Geomatics, University of Ottawa, Ottawa, Ontario, Canada
[6]Thayer School of Engineering, Dartmouth College, Hanover, New Hampshire, USA
[7]Polar Knowledge Canada, Canadian High Arctic Research Station, Cambridge Bay, Nunavut, Canada

**Correspondence:** Ingalise Kindstedt (ingalise.kindstedt@maine.edu)

**Abstract.**

Warming in high alpine regions is leading to an increase in glacier surface melt production, firn temperature, and firn liquid water content, altering regional hydrology and climate records contained in the ice. Here we use field observations and firn modeling to show that although the snowpack at Eclipse Icefield at 3,000 m a.s.l. in the St. Elias Range, Yukon, Canada, remains largely dry, meltwater percolation is likely to increase with an increase in extreme melt events associated with continued atmospheric warming. In particular, the development of year-round deep temperate firn at Eclipse is promoted by an increase in extreme individual melt events, rather than a greater number of small melt events or a prolonged melt season. Borehole temperatures indicate that from 2016 to 2023 there has been a 1.67°C warming of the firn at 14 m depth (to –3.37 ± 0.01°C in 2023). Results from the Community Firn Model show that warming of the firn below 10 m depth may continue over the next decade, with a 2 % chance of becoming temperate year-round at 15 m depth by 2033, even without continued atmospheric warming. Model results also show that the chance of Eclipse developing year-round temperate firn at 15 m depth by 2033 increases from 2 % with 0.1°C atmospheric warming over the period 2023–2033 to 12 % with 0.2°C warming, 51 % with 0.5°C warming and 98 % with 1°C warming. As the majority of the St. Elias Range's glacierized terrain lies below Eclipse, the development of temperate firn at this elevation would represent the ability for widespread meltwater percolation in this region and a wholesale change in its hydrological system, reducing its capacity to buffer runoff and severely limiting potential ice core sites.





## 1 Introduction

Firn is defined as snow that has survived one melt season, and is found in the accumulation region of glaciers, ice caps and ice
sheets around the world (Miller, 1952). There is increasing evidence that firn columns are warming and melting (Vandecrux
et al., 2024; Horlings et al., 2022; Ochwat et al., 2021; Polashenski et al., 2014; Bezeau et al., 2013), which can alter the snow
and firn structure, temperature and chemistry, compromising the climate record in the developing ice column (Samimi and
Marshall, 2017). Liquid water in the snow and firn can also preclude ice core drilling efforts. In addition, firn water retention
can dramatically delay the release of glacier melt to downstream hydrological systems and act as a buffer to glacier runoff
(Culberg et al., 2021; MacFerrin et al., 2019; Koenig et al., 2014; Harper et al., 2012). When deep percolation and localized
freezing occur, firn storage of surface melt can cause a lag in runoff of years to decades (Culberg et al., 2021). Alternatively,
the development of near-surface low-permeability ice layers can lead to increased runoff in the longer term (MacFerrin et al.,
2019; Machguth et al., 2016). Surface melt can be retained in the firn pack, either refrozen in ice layers or in liquid form
as irreducible saturation or a firn aquifer. Firn aquifers account for much of observed firn water storage and can retain water
for several years, both delaying runoff and warming the firn (Ochwat et al., 2021; Miège et al., 2016; Jansson et al., 2003;
Schneider, 1999; Fountain, 1989).

Glacierized high-alpine regions contain unique regional climate records and act as water towers for downstream human
and ecological communities (Miller et al., 2021; Immerzeel et al., 2019; Winski et al., 2018; Fisher et al., 2004). Despite
the importance of warming high-alpine firn columns, comparatively little work has been done on the thermal evolution and
meltwater retention of firn in these regions (e.g. Ochwat et al., 2021; Schneider, 1999; Fountain, 1989) compared to Greenland
(e.g. Amory et al., 2024; Horlings et al., 2022; Culberg et al., 2021; MacFerrin et al., 2019; Machguth et al., 2016; Miège et al.,
2016; Harper et al., 2012). Amplified warming at high elevations (Williamson et al., 2020), combined with high annual snow
accumulation (Marcus and Ragle, 1970) that can insulate underlying firn, make the heavily glacierized St. Elias Range a region
of potentially widespread meltwater retention and active firn aquifer development.

The St. Elias Range (henceforth referred to as "St. Elias") is a mountain range located in the southwest Yukon, Canada, on
the border between the Yukon and Alaska (Fig. 1). Prior work on the upper Kaskawulsh Glacier, a prominent glacier on the
east side of the range, revealed a firn aquifer that may have developed within the last decade, though its age cannot be well
constrained (Ochwat et al., 2021). In contrast, observations from 2002, 2016 and 2017 at Eclipse Icefield ("Eclipse"), ~350
m higher in elevation than the Kaskawulsh aquifer, have shown a relatively dry snow and firn pack (Kochtitzky et al., 2020;
McConnell, 2019; Yalcin et al., 2006). Firn observations that span twenty years, nearby weather station data, and observed
firn aquifer development at lower-elevation sites in the region make Eclipse a compelling case study for firn evolution and
aquifer development in the St. Elias and in glacierized alpine areas more generally. Here we combine field observations and
firn modeling at Eclipse to determine the conditions required to develop a temperate firn column in a formerly polythermal
high-alpine regime. Specifically, we address the following questions: (1) is there evidence for current and increasing melt
production and percolation at Eclipse? (2) how much warming is required to develop year-round temperate firn at Eclipse? (3)
does the distribution of warming throughout the year and/or among intense heat events matter?



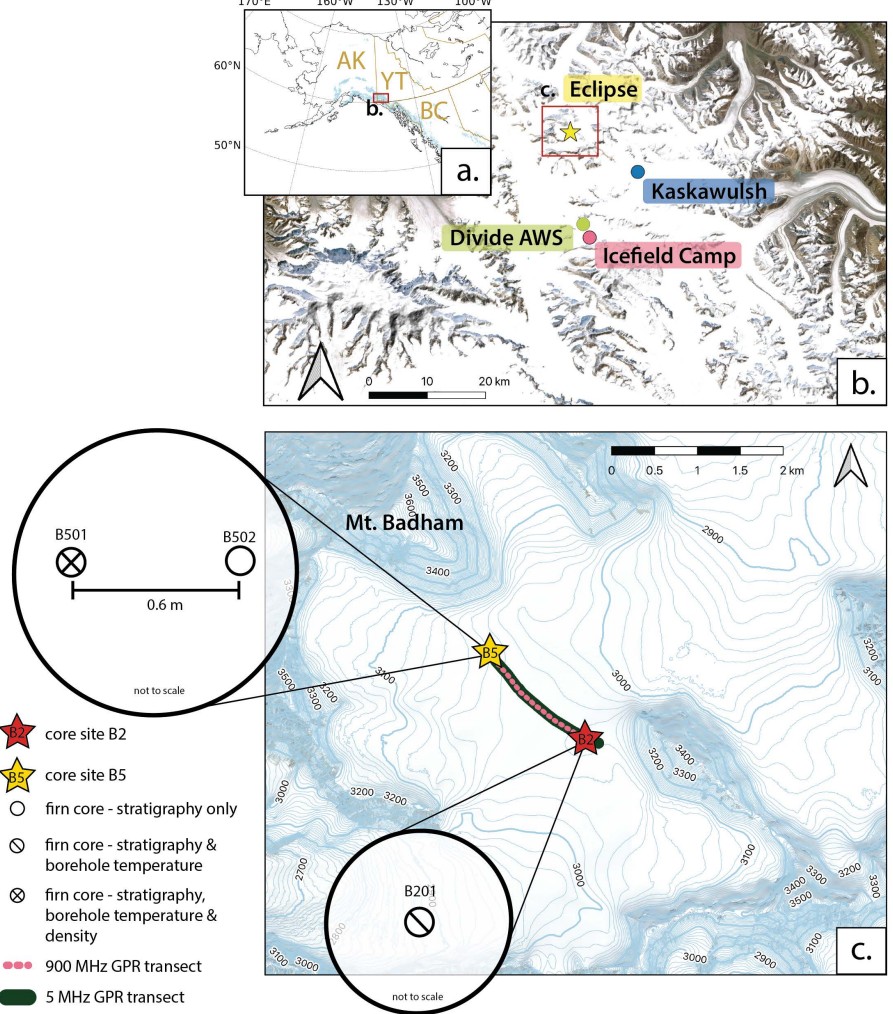

**Figure 1.** Study area in the St. Elias Range with firn cores and ground-penetrating radar collected in the 2023 field campaign. The Gulf of Alaska region is shown in panel (a) with Alaska (AK), Yukon Territory (YT) and British Columbia (BC) labeled in gold. Relevant sites in the St. Elias Range are shown in panel (b), the extent of which is indicated by the red box in panel (a). Panel (c) shows a closeup view of Eclipse Icefield, bounded by the red box marked "Eclipse" in panel (b). Elevation contours in panel (c) are given in meters. Base imagery in panels (b) and (c) is from ESRI.

## 2 Methods

### 2.1 Firn core recovery

We recovered three firn cores from Eclipse Icefield between 2 June 2023 and 4 June 2023 (Fig. 1c). Cores B501 and B502 were
drilled from the surface to depths of 16.2 m and 10.9 m respectively at 60.844°N, 139.851°W, 0.6 m apart from each other to



assess the small-scale spatial variability of near-surface melt features. Core B201 was drilled from the surface to 14.7 m depth at 60.835°N, 139.830°W (at a similar elevation ∼1.5 km to the southeast). Cores B501 and B502 were drilled in one night over the course of five hours; core B201 was drilled the following night. A summary of core recovery information is given in Table 1.

**Table 1.** Firn core recovery information.

| Core | Location | Surface elevation (m a.s.l.) | Recovery date | Recovery start time (local time) | Bottom depth below the snow surface at the time of coring (m) |
|------|----------|------------------------------|---------------|----------------------------------|--------------------------------------------------------------|
| Ecl23_B501 | 60.844°N, 139.851°W | 3,053 | 2 June 2023 | 21:00 (night 1) | 15.6 |
| Ecl23_B502 | 60.844°N, 139.851°W | 3,053 | 3 June 2023 | 00:30 (night 1) | 10.3 |
| Ecl23_B201 | 60.835°N, 139.830°W | 3,058 | 3-4 June 2023 | 22:00 (night 2) | 14.1 |

### 2.1.1 Stratigraphy

We recorded the stratigraphic character of each layer within each core (e.g. opacity, texture, ice layers) in the field and logged accompanying backlit photos. Uncertainty in our stratigraphy comes from the accuracy of the ruler used to measure core depths, as well as the subjective delineation of stratigraphic boundaries within the core. The ruler used was marked to the millimeter, so we report individual layer thicknesses to 0.001 m. However, we report layer depths and thicknesses of stratigraphic sections spanning more than one core segment to 0.01 m to account for the compounded uncertainty over accumulating core segments. We report all layer depths relative to the last summer surface (LSS) because of high uncertainty in our measurements of seasonal snow depth resulting from the visual similarity of seasonal snow and core chips (a drilling artifact) pulled out of the drill barrel. For visual representation purposes, we show the LSS to be located at ∼4.25 m depth, within the range of plausible LSS depths based on our firn core observations (4.0–4.5 m).

### 2.1.2 Density

We recorded density in the field for core B501 starting at 1.92 m depth. We did not record density for the unconsolidated snow above this because the volume uncertainty of core segments was too high. Core sections were sawed into roughly 10 cm segments, or delineated at existing breaks. We then measured each segment's length, diameter and mass, and assessed its cylindrical completeness. Following the methods of Ochwat et al. (2021), we assigned each core segment a cylindrical completeness value ($f$) of 0–1, with 1 denoting an intact core and fractional values denoting the portion of core segment



present. For example, a core segment determined via visual inspection to have 5 % of its volume missing due to chipping or crumbling would be assigned an $f$ value of 0.95. Density was calculated for each core segment using:

$$\rho = m/V, \qquad with \qquad V = f\pi L(D/2)^2 \tag{1}$$

where m is mass, $V$ is volume, $L$ is core segment length, and $D$ is the average diameter. We removed outliers from the dataset
if they were not physically plausible (i.e. either $> 917$ kgm$^{-3}$ or $< 300$ kgm$^{-3}$ below the density of the last summer surface within uncertainty). We report density for the top depth of each segment. For example, the density value reported for a depth of 3.40 m would be measured over the segment from 3.40 to 3.50 m depth below the snow surface at the time of coring.

Uncertainty in our density measurements comes from a variety of sources and can be calculated based on the uncertainties in mass and volume with the following:

$$d\rho = \rho\sqrt{\left(\frac{dm}{m}\right)^2 + \left(\frac{dV}{V}\right)^2} \tag{2}$$

Mass uncertainty arises from the accuracy of the scale. The scale used has a nominal accuracy of $\pm 0.1$ g, but we assign a mass uncertainty of 0.3 g to account for residual snow or water on its surface. Volume uncertainty comprises uncertainty in the measured segment length and radius (used to calculate cross-sectional area $A$) and in the subjective assignment of cylindrical completeness ($f$):

$$dV = V\sqrt{\left(\frac{df}{f}\right)^2 + \left(\frac{dA}{A}\right)^2 + \left(\frac{dL}{L}\right)^2} \tag{3}$$

Our ruler was marked every millimeter, but following Ochwat et al. (2021), we assign $dL = dD = 0.25$cm to account for rough-cut core segments with crumbly edges. We then calculate the uncertainty in area using:

$$dA = \pi D\left(\frac{dD}{2}\right) \tag{4}$$

We also follow Ochwat et al. (2021) in assigning an uncertainty of $df = 0.1$ for $f \geq 0.8$ and $df = 0.2$ for $f < 0.8$, because a
less complete core is more difficult to visually assess. We calculate the uncertainty for mean density values from the root mean square of all point values being averaged:

$$d\rho_k : d\overline{\rho} = \frac{1}{N}\left[\sum_N d\rho_k^2\right]^{1/2} \tag{5}$$

We report the top depth of all density measurements relative to the snow surface assuming an LSS depth of 4.24 m below the surface. Consequently, absolute depths reported have an uncertainty of $\pm 0.2$ m; however, density calculations depend on
segment thickness rather than absolute depth and therefore use the assigned measurement uncertainty of 0.25 cm.





### 2.1.3   Borehole temperatures

We obtained temperature measurements from the boreholes of cores B501 and B201 using an RBRsolo³ T compact single-channel temperature logger. We allowed the borehole to equilibrate for approximately 12 hours after drilling before taking any temperature measurements. We obtained one temperature measurement near the bottom of B201 (14.1 m below the snow surface) after the sensor was allowed to equilibrate in the borehole for 1.5 hours, and one temperature measurement near the bottom of B501 (15.5 m below the snow surface) after the sensor was allowed to equilibrate in the borehole for 1.5 hours. In addition to these deep equilibrated measurements, we acquired a full-length temperature profile for each core. Due to time constraints in the field, this was based on 30 s of measurement time at each depth, after 15 s of equilibration time. After recording the surface air temperature, we lowered the logger into the borehole using a survey wheel while continuously sampling at a rate of 2 Hz. At 20 cm intervals, we paused lowering and recorded a timestamp to identify the temperature at that depth. RBRsolo³ T temperature measurements have an initial accuracy of ± 0.002°C, which we apply to our basal borehole temperatures when the sensor was allowed to equilibrate in the borehole for 1.5 hours. We assign an uncertainty of 0.01°C to our temperature profiles at 20 cm increments when a 15 s equilibrium time was used.

## 2.2   Ground-penetrating radar

We collected ground-penetrating radar (GPR) data at frequencies of 900 MHz and 5 MHz, using two different systems to image the near subsurface (∼20 m depth) and full ice thickness (up to 700 m depth) respectively. We collected very high frequency (VHF) data on 2 June 2023 with a GSSI system, using a shielded 900 MHz center frequency antenna (model 3101) and an SIR4000 control unit paired with a Garmin GPSMap78 (horizontal accuracy of ± 10 m, vertical accuracy of ± 3 m). We collected data at 2,048 samples per scan and 24 scans per second with a range of 240 ns. We collected high frequency (HF) data on 4 June 2023 with a Blue Systems Integrated (BSI) 5 MHz antenna, with 30 m separation between the transmitter and receiver. We ski-towed both GPR systems at a rate of ∼1 ms⁻¹ to obtain profiles of stratigraphy between core sites B2 and B5 (Fig 1c). Radar data were processed in ImpDAR (Lilien et al., 2020) using standard processing techniques, including: clipping stationary periods, constant trace spacing, bandpass filtering, and normal move out analysis to correct for antenna separation. HF data were also migrated using the SeisUnix sumigtk routine to correct for hyperbolic reflections from basal topography. Depth-variable density profiles obtained from shallow (2023 B501) and deep (2002) cores were used to calculate variable, permittivity-dependent, radar-wave velocities to determine the depth of reflected layers. The LSS was semi-automatically picked across the VHF profile.

## 2.3   Firn modeling

We model the firn pack evolution over a period of 20 years (2013–2033) at Eclipse using the Community Firn Model (CFM), an open-source, modular model framework coded in Python 3 and available for download on Github (Stevens et al., 2020). We used the CFM to simulate firn density and temperature evolution as well as meltwater retention and refreezing. We parameterize firn density evolution using the densification scheme from Kuipers Munneke et al. (2015) and an assigned surface density. The



**Table 2.** Suite of degree day factors (DDFs) and surface density values tested in model tuning. A total of 300 DDF and surface density combinations were tested.

| Variable | Values | | Source |
|---|---|---|---|
| DDF (mm°C$^{-1}$d$^{-1}$) | 2.6 | 5.8 | MacDougall et al. 2011: minimum (2.6 mm °C$^{-1}$d$^{-1}$) and maximum (8.2 mm |
| | 3.0 | 6.2 | °C$^{-1}$d$^{-1}$) values derived from 2008 and 2009 data on two glaciers in the |
| | 3.4 | 6.6 | Donjek Range*. The glaciers are located between 60.783°N and 60.950°N, |
| | 3.8 | 7.0 | and 139.083°W and 139.217°W, and range from 1,890–3,100 ma.s.l. in |
| | 4.2 | 7.4 | elevation. |
| | 4.6 | 7.8 | |
| | 5.0 | 8.2 | |
| | 5.4 | | |
| Surface density (kgm$^{-3}$) | 225 | 430 | McConnell 2019: surface density measured in Eclipse 2016 core (430 kgm$^{-3}$) |
| | 250 | 440 | |
| | 275 | 450 | Ochwat et al. 2021: seasonal snow density measured in Kaskawulsh 2018 |
| | 300 | 460 | cores (450 kgm$^{-3}$). The drill site was located at 60.78° N, 139.63° W at an |
| | 325 | 470 | elevation of 2,640 m a.s.l. |
| | 350 | 480 | |
| | 375 | 490 | Pulwicki et al. 2018: minimum (227 kgm$^{-3}$) and maximum (431 kgm$^{-3}$) |
| | 400 | 500 | measured surface density values over three glaciers in the Donjek Range*. The |
| | 410 | 510 | glaciers were located between 60.791° N and 60.992° N, and 139.079° W and |
| | 420 | 520 | 139.246° W, covering an elevation range from 1,899–3,103 m a.s.l. |

*a subrange in the St. Elias

CFM uses a bucket scheme to simulate meltwater percolation, and it uses an enthalpy-based heat transfer scheme to simulate heat diffusion in the presence of phase changes. We use a parameterization for thermal conductivity from Calonne et al. (2019).
We force the model with air temperature data from a weather station near the ice divide between the Kaskawulsh and Hubbard Glaciers (30 km away, "Divide AWS"; Fig. 1b), surface melt calculated from air temperatures using a simple degree day model, and mean annual accumulation rate of 1.4 mw.e.a$^{-1}$ (McConnell, 2019) distributed evenly throughout the year. We elevation-correct Divide AWS air temperatures to Eclipse (∼400 m higher) using a lapse rate of –3.98°Ckm$^{-1}$ following Hill et al. (2021).
We explore the model's sensitivity to 15 different degree day factor (DDF) values used to estimate surface melt from air temperatures and 20 different surface density values (Table 2). Our range of DDF values (2.6–8.2 mm°C$^{-1}$d$^{-1}$) is bounded by the minimum and maximum DDFs derived from 2008–2009 in situ data from two glaciers on the northeast side of the St. Elias Range (MacDougall et al., 2011). Our tested surface density values span the range from 225–520 kgm$^{-3}$, covering the full range of surface snow densities measured at Eclipse (McConnell, 2019), at two sites near the Kaskawulsh/Hubbard





Divide (Ochwat et al., 2021; McConnell, 2019), and over three glaciers in the nearby Donjek Range (Pulwicki et al., 2018). Locations and elevations of measured regional surface densities are found in Table 2. We test a higher concentration of surface density values from 400–520 $\text{kgm}^{-3}$ since in situ data suggest this is the most reasonable range of surface density estimates for Eclipse. We select a representative pairing (DDF = 6.2, $\rho$ = 450 $\text{kgm}^{-3}$) from all the combinations of DDF and surface density values that produce no liquid water down to 14 m depth in the firn in both spring 2016 and spring 2023 (consistent

with firn cores and GPR showing no evidence of liquid water at those times), and a firn temperature between –2°C and –4°C at 14 m depth in spring 2023 (consistent with 2023 borehole temperature measurements) to predict the evolution of the firn pack from 2024–2033. Our exploration of model sensitivity to DDF and surface density values is detailed in Appendix A.

We spin the model up from ∼1983–2013 (exact spinup time varies slightly among model runs as it is dependent on densi-fication rate and surface melt) using downscaled North American Regional Reanalysis (NARR) air temperatures for Eclipse

from 1983 to 2013 (Jarosch et al., 2012). We also test the model sensitivity to three other spinup schemes using different air temperature datasets (Appendix A): 1) elevation-adjusted Divide AWS data from 2013 to 2024 repeated for the duration of the spinup, 2) synthetic climate data selected from a Gaussian distribution of temperatures based on elevation-adjusted 2013–2024 Divide AWS data, and 3) like spinup scheme (2) but with a historical 0.024°$\text{Ca}^{-1}$ rate of temperature change applied for the duration of the spinup such that the mean annual temperature at the start of the main model run is consistent with elevation-

adjusted 2013 Divide AWS data. All model spinups are forced with the same mean annual accumulation rate (1.4 $\text{mw.e.a}^{-1}$) used in the main model run.

We run the model with approximately one-day timesteps. After spinup (∼1983 to 2013), we apply the elevation-adjusted Divide AWS temperature data as forcing from 2013–2024, and then run the CFM under a suite of climate scenarios from 2024–2033, including: (a) continuation of current climate, (b) 0.1°C cooling, (c) 0.1°C warming, (d) 0.2°C warming, (e)

0.5°C warming, and (f) 1°C warming by 2033. We generate synthetic air temperatures for all 2024–2033 climate scenarios by assigning daily temperature values randomly drawn from a Gaussian distribution of temperatures based on elevation-adjusted 2013–2024 Divide AWS data. Temperatures for scenario (a) are drawn from the distribution described by the mean and standard deviation of 2013–2024 data. Temperatures for scenarios (b-f) are drawn from distributions with a prescribed rate of change applied to the mean such that the specified degree of warming or cooling for each scenario is reached by December 31, 2033.

For each climate scenario, we run the model fifty times and calculate the percentage of model runs that produce temperate firn at 15 m depth by 2033.

## 2.4   Comparison among St. Elias study sites

To contextualize the firn conditions at Eclipse, we compare our results with data from two neighboring sites near the ice divide between the Kaskawulsh and Hubbard Glaciers: Icefield Divide Camp (60.68°N, 139.78°W; 2,603 $\text{ma.s.l.}$), which

we refer to as "Icefield Camp" to avoid confusion with reference to the broader Kaskawulsh/Hubbard Divide area, and the upper Kaskawulsh Glacier (60.78°N, 139.63°W; 2,640 $\text{ma.s.l.}$), which we refer to as "Kaskawulsh" (Fig. 1b). In 2018, VHF (400 MHz) and HF (5 MHz) ground-penetrating radar data were collected at Icefield Camp; for a full description of GPR deployment and processing, see McConnell (2019). In the same year, two firn cores were recovered from the upper northern





arm of Kaskawulsh Glacier by Ochwat et al. (2021). The cores were each 8 cm in diameter and drilled 60 cm apart from each
other to depths of 36 m (Core 1) and 21 m (Core 2); liquid water was encountered at 34.5 m below the surface in Core 1
(Ochwat et al., 2021). Stratigraphy of both cores and density of Core 1 were measured in the field; densities for a subset of
samples from Core 2 were measured after transporting the core from the field to nearby Kluane Lake Research Station. For a
complete description of Kaskawulsh core recovery and analysis, see Ochwat et al. (2021).

We also contextualize the Eclipse, Icefield Camp and Kaskawulsh sites within the St. Elias Range by producing a regional
hypsometric curve for the region's glacierized terrain. We derive this curve from the ArcticDEM digital elevation model at 10
m resolution, and highlight areas ranging from 2,600 to 3,000 ma.s.l., which cover the elevation band between the our sites of
interest.

### 2.5 Firn changes over time

The St. Elias Range has been an area of glaciological study for decades. To investigate how firn at Eclipse has changed over
time, we compare our 2023 data to other measurements made at Eclipse since 2002:

(i) We compare 2023 GPR data to both VHF (400 MHz) and HF (10 MHz) GPR data from 2016, described in McConnell
(2019).

(ii) We compare 2023 density measurements to values measured in firn cores drilled in both 2016 and 2017 (McConnell,
2019). Density in the 2016 Eclipse ice core was measured by weighing each core segment (to the nearest 10 g) and dividing
by the volume computed from one length measurement using a ruler ($\pm$ 0.5 cm accuracy) and the average of three diameter
measurements using calipers ($\pm$ 0.5 mm accuracy).

(iii) We compare 2023 borehole temperatures to values measured in 2016 using a string of iButtons and Easylog USB
temperature sensors lowered into a borehole and allowed to equilibrate for over 24 hours before logging data. iButton and
Easylog sensors were both deployed at a sample rate of one per 300 s and both sensors have an instrumental accuracy of 0.5°C.
For depths with multiple sensors, the mean of the temperature readings was taken as the final value.

(iv) We compare 2023 density measurements to values from a 130 m ice core recovered in 2002 (Kochtitzky et al., 2020;
Yalcin et al., 2006). Density values for the 2002 core were calculated in the field using bulk mass and volume measurements
at 1 m increments (Kochtitzky et al., 2020; Kelsey et al., 2012). Ice layers were assumed to have a density of 910 kgm$^{-3}$ and
accounted for in the bulk density calculations.

## 3 Results

### 3.1 Stratigraphy

The stratigraphy of all three 2023 cores shows ice layers, ice lenses, and melt-affected firn throughout the core (Fig. 2). The
first ~4 m of all three cores comprise the seasonal snowpack, with the last summer surface (LSS) appearing as a crusty layer



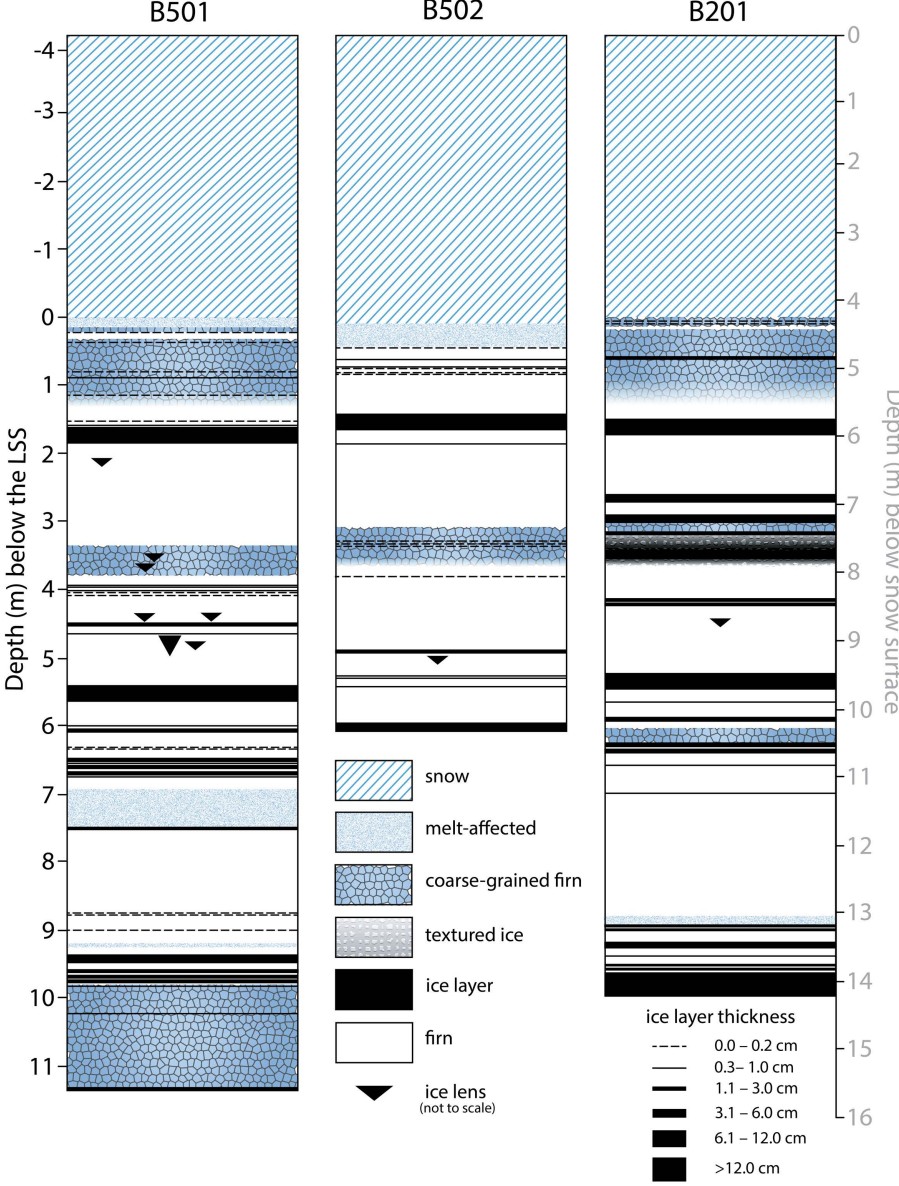

**Figure 2.** Stratigraphy of the three firn cores drilled at Eclipse Icefield in 2023. The position of the LSS 0 mark in the lefthand depth scale is the mean LSS depth (4.27 m).

of melt-affected firn. Unless otherwise specified, all following stratigraphic depths are reported relative to the LSS, which we

designate as below the LSS or "BLSS".

In core B501, most of the top meter of firn below the LSS is coarse-grained and contains several hairline to 1.0 cm ice layers (Fig. 2). We use the term "coarse-grained" to refer to sections of firn composed of coarse (1-2 mm) well-sintered ice grains





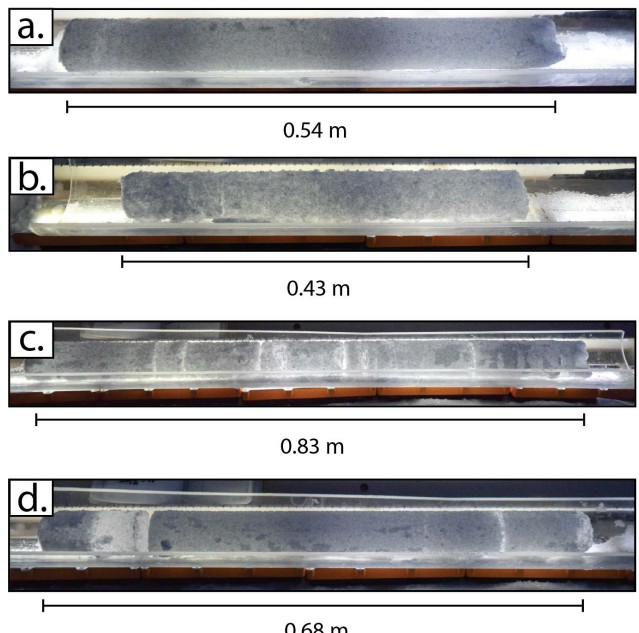

**Figure 3.** Example backlit photos from core B501 of: (a) undisturbed firn (6.97 m BLSS), (b) coarse-grained firn (3.40 m BLSS), (c) clustered thin ice layers (6.36 m BLSS), and (d) a thick (>10 cm) ice layer (5.46 m BLSS). Depths refer to top of the core segment below the last summer surface (BLSS).

(Fig. 3b). Unlike the large, faceted grains in surface and depth hoar, which reduce layer strength, the rounded, well-sintered grains in our coarse-grained firn result in a strong icy layer requiring a saw to cut through. Despite their sintering, the grains do
remain readily distinguishable from one another, differentiating these layers from blue glacier ice. Approximately 1 m BLSS, the firn transitions from coarse-grained and very melt-affected to fine-grained with fewer melt features. It remains fine-grained until 3.40 m BLSS, where there is a sharp transition from fine- to coarse-grained firn. The largest sections of fine-grained firn occur from 2.15 to 3.40 m BLSS (1.25 m thick) and from 7.54 to 8.83 m BLSS (1.29 m thick). The rest of the core contains clusters of ice layers and lenses, or is otherwise melt-affected based on texture and appearance. The thickest ice layer in the
core begins at 5.46 m BLSS and is 12.0 cm thick. Core B501 has a total firn ice content of 5 % by volume and 40 % of its length below the LSS (11.34 m) is visibly metamorphized and/or melt-altered.

In core B502, the first meter of firn below the LSS lacks the large crystals observed in core B501, but does contain numerous hairline to 1.0 cm thick ice layers (Fig. 2). The largest sections of fine-grained firn occur from 1.80 to 3.03 m BLSS (1.23 m thick) and from 3.72 to 4.82 m BLSS (1.10 m thick). The thickest ice layer in core B502 begins at 1.41 m BLSS and is 10.0
cm thick. Core B502 has a total ice content of 3 % by volume and 19 % of its length below the LSS (5.96 m) is melt-altered.

In core B201, the LSS is defined based on the first appearance of coarse-grained firn rather than a crust as in B501 and B502 (Fig. 2). The top meter of firn below the LSS is coarse-grained and contains several ice lenses. At 1.18 m BLSS, the firn transitions from coarse- to fine-grained. The largest sections of fine-grained firn occur from 1.70 to 2.71 m BLSS (1.01 m





thick) and from 7.10 to 8.90 m BLSS (1.80 m thick). Core B201 has three ice layers over 10 cm thick, the largest being at least
33 cm, where we stopped drilling because of mechanical difficulties. Core B201 has a total ice content of 10 % by volume and
27 % of its length below the LSS (9.91 m) is melt-altered.

Ice content and melt alteration of cores B501 and B201 from the LSS down to 14.15 m depth (below snow surface) are
summarized in Table 3. Although all three cores differ in their ice content and layer characteristics, we focus on the difference
between cores B501 and B201 to maximize the depth of overlap for a greater sample domain.

**Table 3.** Ice content and melt alteration of cores B501 and B201 from the LSS to 14.15 m depth below the snow surface.

| core | % ice content by vol. | % of ice layers >1 cm thick by vol. | % of total ice content in thick layers | % melt features by vol. |
|------|----------------------|-------------------------------------|----------------------------------------|-------------------------|
| B501 | 3 % | 24 % | 82 % | 17 % |
| B201 | 10 % | 48 % | 93 % | 27 % |

## 3.2 Density

Density measurements for core B501 are shown in Figure 4. We report density values beginning at 1.92 m depth (below the
snow surface) because of high uncertainty in our volume measurements for the surface snow. The mean density from 1.92 m
to the LSS at 4.24 m depth is $612 \pm 20$ kgm$^{-3}$. Seasonal snow above the LSS was dry with no ice content. We focus here on
the firn below the LSS, much of which shows signs of melt alteration. The mean density of the top 2 m of firn below the LSS
is $638 \pm 21$ kgm$^{-3}$. The mean density of the top 10 m of firn below the LSS is $689 \pm 10$ kgm$^{-3}$. The mean density of the
bottom 2 m of the core is $722 \pm 23$ kgm$^{-3}$. The mean overall density of core B501 is $679 \pm 9$ kgm$^{-3}$. In general, density
increases with depth throughout the core. However, cyclic variations can be seen, which are likely seasonal, particularly in the
top 10 m. Individual ice layers can also be identified by peaks in density. Note that although measured densities for these ice
layers are implausibly high, their values are physically reasonable within uncertainty.

## 3.3 Borehole temperatures

We obtained borehole temperatures down to 14.1 m depth (below the snow surface) for B201 and to 15.5 m depth (below the
snow surface) for B501 (Fig. 5). Both profiles show an initial cooling for the top ~3 m, which then transitions to a warming,
with a temperature minimum just below –8°C in the top 4 m of the profile. In B201, the warming continues until ~12 m depth,
below which the profile shows a temperature stabilization and slight cooling. In B501, the warming that begins around 3 m
depth continues through the bottom of the profile domain (15.5 m). At 14 m depth, B2's temperature is –1.74 ± 0.01°C and
B5's temperature is –3.37 ± 0.01°C. Although the temperature probe was continuing to equilibrate in the borehole after 45
seconds time (15 s equilibration and 30 s measurement), our fully equilibrated spot measurements (indicated by stars in Figure
5) are consistent with measurements acquired after only 15 seconds equilibration. We therefore consider our full temperature
profiles to be adequately representative of borehole conditions.



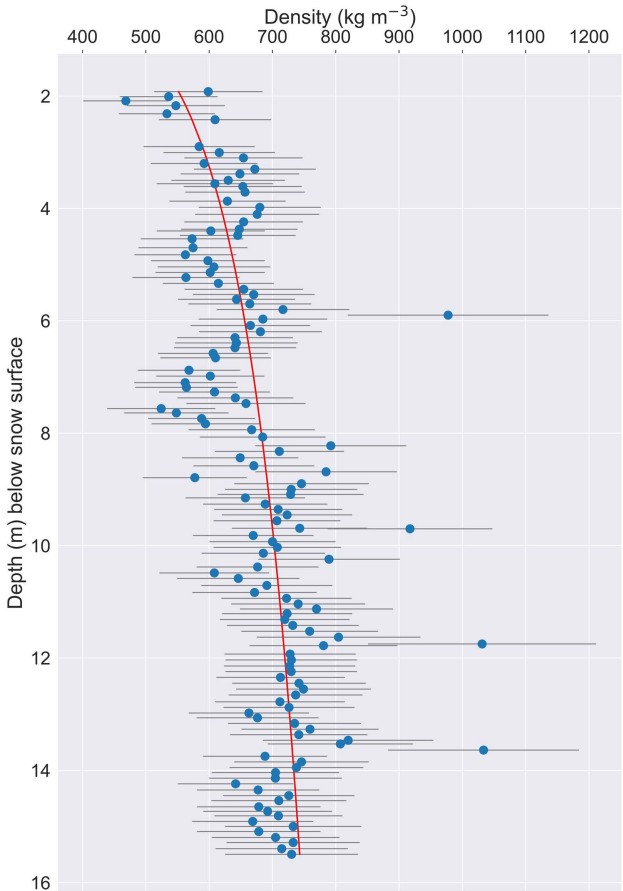

**Figure 4.** Firn densities of 2023 core B501 with uncertainty values (gray) and a best-fit logarithmic curve (red). Measured densities begin at 1.92 m because the snow above that depth was too crumbly for accurate volume measurements.

## 3.4 Ground-penetrating radar

Clear, continuous englacial horizons are observable throughout the radargrams, which is generally indicative of a non-temperate snowpack (Fig. 6). For example, our VHF (900 MHz) radargram shows a continuous reflector between 3 and 4 m depth (Fig. 6b), which we interpret to be the LSS. The LSS is determined to be at 3.66 m at site B2 and 3.60 m at site B5, differing from the average LSS depth observed in our firn cores because of assumptions about the radio wave velocity used to interpret the GPR data. Similarly, our HF (5 MHz) radargram shows a continuous reflector at ∼150 m depth, which we interpret to be the ash layer associated with the Katmai eruption of 1912 (Fig. 6c). Our HF radargram also shows the total ice thickness, which ranges from ∼150 m to greater than 700 m (Fig. 6c). The deepest ice is found near the center of the profile, with the bedrock sloping up toward either end. Bedrock is located 484 m below the surface at site B2 and 566 m below the surface at site B5. There is no evidence for a firn aquifer in any of our GPR data.





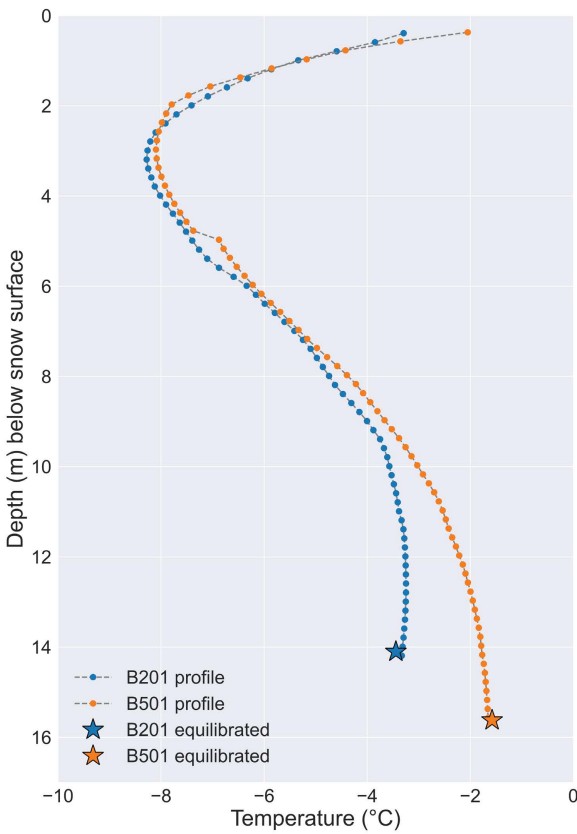

**Figure 5.** Borehole temperatures measured at Eclipse in 2023. Full 2023 profiles (circles) were acquired with 15 s sensor equilibration at each depth. Near the bottom of each borehole, one measurement (star marker) was acquired after allowing the sensor to equilibrate for 1.5 hours.

## 265 3.5 Firn modeling

Assuming a continuation of current climate conditions through 2033, our reference model predicts little change in firn temperature below 10 m depth; over fifty replicate runs, our median predicted firn temperature is –2.90°C at 10 m depth, –2.64°C at 15 m depth, and –2.32°C at 20 m depth in 2033. However, our range of predicted firn temperatures is 6.05°C at 10 m depth, 5.11°C at 15 m depth, and 4.74°C at 20 m depth, with the reference model producing temperate firn for at least some of the 270 fifty replicate runs at each depth (Fig. 7). Eclipse therefore appears to be near a threshold for supporting temperate firn below the penetration depth (∼10 m) of the annual temperature wave.

Additionally, firn conditions (even below 10 m depth) at Eclipse are sensitive to changes in air temperature and exhibit threshold behavior with projected warming (Fig. 8). Over 50 replicate model runs with 0.1°C cooling by 2033, firn at 15 m depth has a 0 % chance of being temperate year-round in 2033, compared to a 2 % chance over 50 replicate runs under a





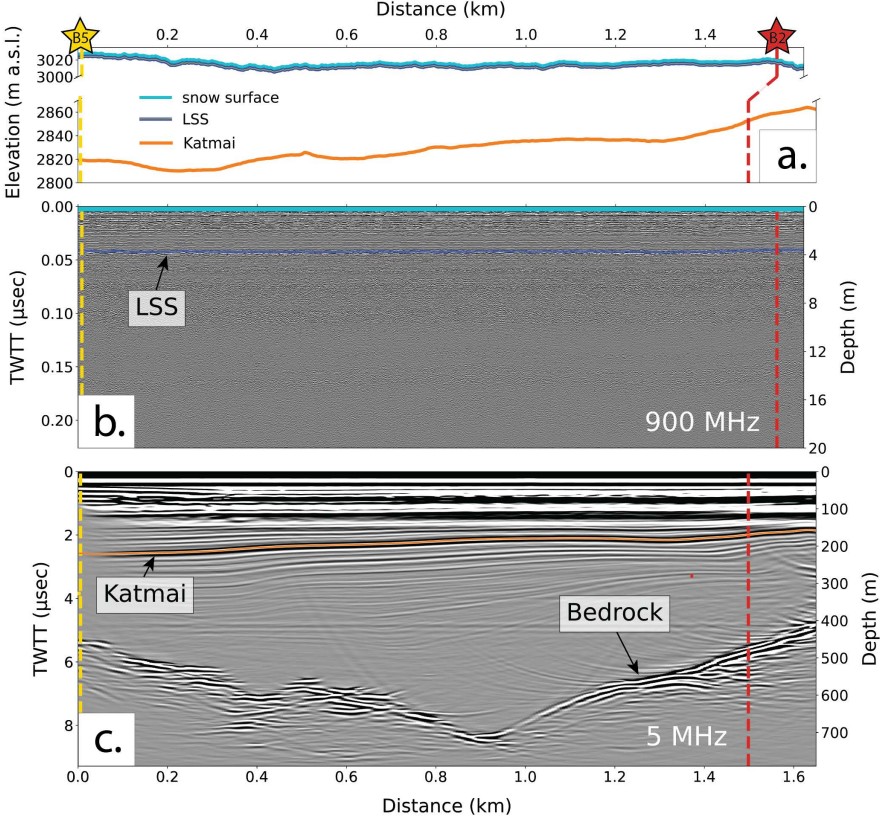

**Figure 6.** Ground-penetrating radar data showing the last summer surface (LSS), Katmai volcanic ash layer and bedrock across a transect between core sites B5 and B2. Panel (a) shows surface, LSS and Katmai layer elevations; panels (b) and (c) are radargrams from the 900 MHz and 5 MHz systems respectively. Yellow and red dashed lines indicate the surface location of core sites B5 and B2 respectively. The x-axes of panels (a-c) are all on the same distance scale; however, the core site locations vary slightly between the shallow (900 MHz) and deep (5 MHz) profiles because the transects skied with the two systems were not exactly aligned.

continuation of current climate or with 0.1°C warming. The probability of firn at 15 m depth becoming temperate year-round increases to 12 % with 0.2°C warming, 51 % with 0.5°C warming and 98 % with 1°C warming by 2033. The evolution of density, liquid water content and temperature over the full firn column is shown for a reference model run under each modeled climate scenario in Fig. 9.

        To examine the conditions associated with the production of year-round temperate firn at 15 m depth, we focus on model
runs with 0.2°C and 0.5°C warming by 2033. We select these two climate scenarios because all others have a sample size of n ≤ 1 for either model runs that produce temperate firn by 2033 or those that don't. Conditions associated with the development of year-round temperate firn at 15 m depth include higher total melt season positive degree days (PDDs) and greater magnitude



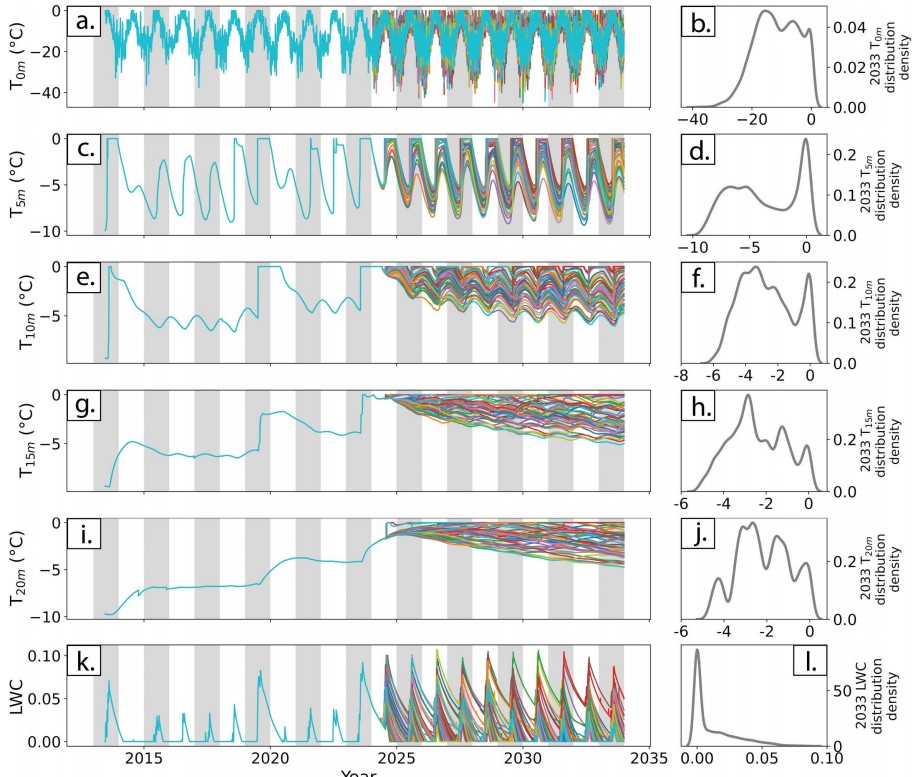

**Figure 7.** Firn temperature (a-j) and liquid water content (LWC; k-l) timeseries and distribution plots for replicate runs of our reference model at Eclipse Icefield. Temperatures are shown at 0 m (a-b), 5 m (c-d), 10 m (e-f), 15 m (g-h), and 20 m (i-j) depth. LWC is calculated for the entire firn column. A degree day factor (DDF) of 6.2 and a surface density of 450 $\text{kgm}^{-3}$ were used for all fifty model runs. All model runs were forced using downscaled NARR air temperatures during the spinup period (pre-2013), elevation-corrected Divide AWS data from 2013–2024, and temperatures randomly drawn from a Gaussian distribution described by the mean and standard deviation of elevation-corrected 2013–2024 Divide AWS temperatures from 2024–2033. Forcing temperatures from 2024–2033 assume a continuation of current climate conditions. Gray bars show odd-numbered years (e.g. 2015, 2017, etc.). Values on the x-axis of the righthand panels (distributions) correspond to those on the y-axis of the lefthand panels (timeseries). Distributions are shown for 2033 outputs only.

of individual melt events. We define melt events as any period over which the snow surface is continuously melting without refreeze.

For model runs that do not develop year-round temperate firn at 15 m depth by 2033, the median of the total melt season PDDs (over all model runs for any given year) never exceeds 36 (Fig. 10, Table B1). In contrast, the median total melt season PDDs over all model runs that do produce temperate firn by 2033 ranges from 36.68 in 2024 to 55.24 in 2033. We use the months of May through September to represent the typical melt season in these reported totals.

Additionally, we find a significant ($p < 0.05$) difference (Wilcoxon Rank Sum test) between the median melt event magnitude
(mm) for model runs that produce year-round temperate firn at 15 m depth and those that don't under both 0.2°C warming and





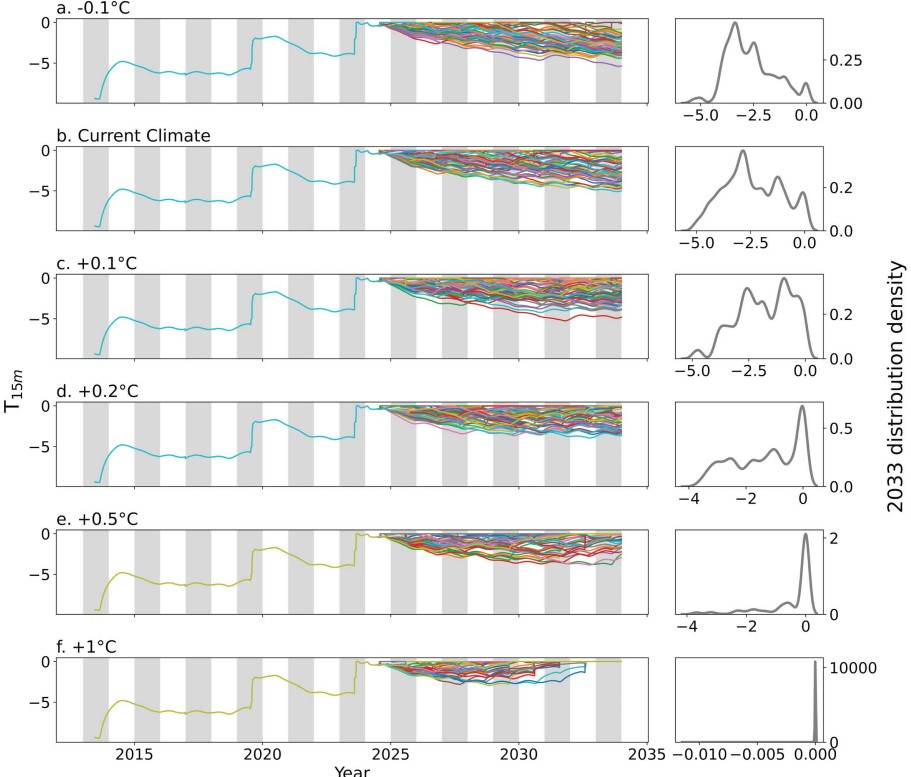

**Figure 8.** 2013–2033 timeseries and 2033 distribution plots of firn temperature at 15 m depth. All model runs shown use a degree day factor of 6.2 and a surface density of 450 kgm$^{-3}$. All model runs were forced using downscaled NARR air temperatures during the spinup period (pre-2013), elevation-corrected Divide AWS data from 2013–2024, and temperatures randomly drawn from a Gaussian distribution described by the mean and standard deviation of elevation-corrected 2013–2024 Divide AWS temperatures from 2024–2033.

0.5°C warming. We do not find any significant ($p < 0.05$) difference in the number of melt events. We also see no significant ($p < 0.05$) difference in median melt season start, end, or length. Complete results of our Wilcoxon Rank Sum tests are shown in Appendix B (Table B2).

## 3.6 Comparison among St. Elias study sites

Over 80 % of the St. Elias Range's glacier cover lies below both Eclipse and the Kaskawulsh/Hubbard Divide in elevation (Fig. 11). Although it is only about 400 m higher in elevation than Divide, the firn pack at Eclipse to date remains drier than that at both Icefield Camp and Kaskawulsh study sites. Comparisons between VHF (400 MHz) and HF (5-10 MHz) GPR data from Eclipse and Icefield Camp in 2016 and 2018, respectively, showed stratigraphic differences indicating a wetter firn pack at Icefield Camp relative to Eclipse, including a bright reflector in the HF data at ∼25 m depth suggesting the presence of a

liquid water table in the firn at Icefield Camp but not at Eclipse (McConnell, 2019). VHF data from Icefield Camp show greater



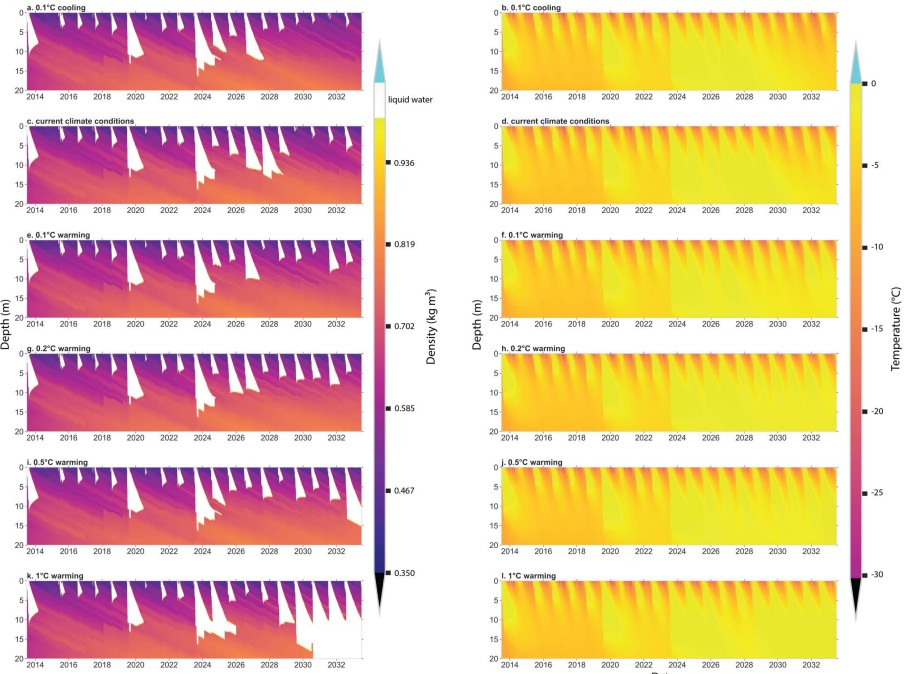

**Figure 9.** Timeseries of density, liquid water content and firn temperature from 2013–2033 under six different climate scenarios: 0.1°C cooling by 2033 (a-b), continuation of current climate through 2033 (c-d), 0.1°C warming by 2033 (e-f), 0.2°C warming by 2033 (g-h), 0.5°C warming by 2033 (i-j), and 1°C warming by 2033 (k-l). The righthand panels show both density and LWC, with density shown by the colorbar and white areas indicating the presence of liquid water. All climate scenarios are prescribed using surface temperature values drawn from a Gaussian distribution based on elevation-corrected 2013–2024 AWS data from Divide with the appropriate level of warming or cooling applied.

signal attenuation than those from Eclipse, consistent with a wetter firn pack at Icefield Camp (McConnell, 2019). GPR data from Eclipse in 2023 show the same results: clear stratigraphy with low signal attenuation, and a lack of bright reflector that would indicate a water table.

At the Kaskawulsh site, two firn cores were drilled in 2018 that allow comparison of firn properties between Kaskawulsh
and Eclipse. The Kaskawulsh cores showed extensive evidence of meltwater percolation and freezing events, including the saturation of firn with liquid water below 34.5 m depth (Ochwat et al., 2021). The saturated firn, or firn aquifer, likely developed since 2013 based on model results, but its age cannot be confirmed (Ochwat et al., 2021). The seasonal snow layers at both Eclipse in 2023 and on the Kaskawulsh in 2018 were dry since drilling at both sites occurred in the very early stages of the melt season. Ice content in the Kaskawulsh 2018 Core 1 is similar to that in the Eclipse 2023 core B201, and less than Eclipse
2023 cores B501 and B502 (Table 4). However, the density of the upper 10 m of firn (below the LSS) was higher in Eclipse core B501 ($688 \pm 10$ kgm$^{-3}$) than in the Kaskawulsh 2018 cores ($588 \pm 8$ kgm$^{-3}$ and $572 \pm 7$ kgm$^{-3}$; Ochwat et al., 2021).



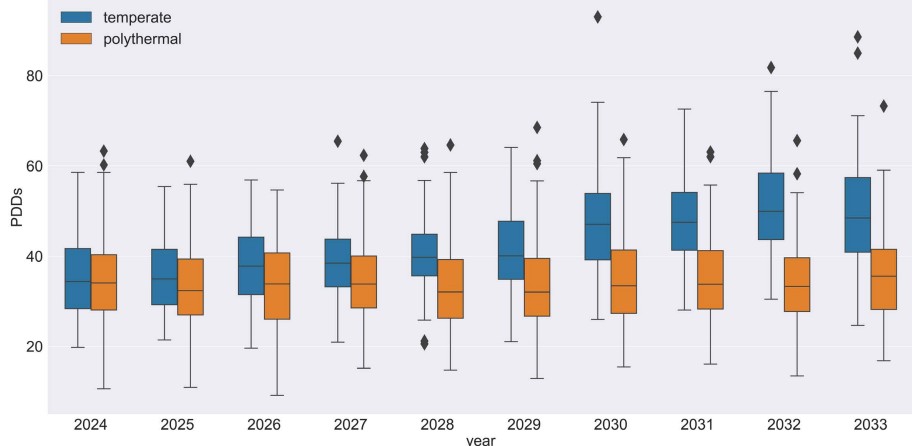

**Figure 10.** Positive degree days (PDDs) from May through September of each year for model runs that produce year-round temperate firn at 15 m depth by 2033 (blue) and those that don't (orange).

Much like cores B2 and B5 at Eclipse, Cores 1 and 2 on the Kaskawulsh showed some similarities in general regions of melt alteration but distinct individual stratigraphic layers.

**Table 4.** Ice content of Eclipse 2023 cores and Kaskawulsh Core 1. Kaskawulsh data are from Ochwat et al. (2021).

| Core | Total length (m) | Total length ice (m) | Total ice content by vol. |
|---|---|---|---|
| Eclipse 2023 B201 | 14.15 | $1.03 \pm 0.03$ | 7.3 % |
| Eclipse 2023 B501 | 15.58 | $0.54 \pm 0.04$ | 3.5 % |
| Eclipse 2023 B502 | 10.28 | $0.19 \pm 0.02$ | 1.8 % |
| Kaskawulsh 2018 Core 1 | 36.6 | $2.33 \pm 0.26$ | 7.2 % |

## 3.7 Firn changes over time

A borehole temperature of $-3.37 \pm 0.01°$C was recorded at 14 m depth in May 2023 at site B2, approximately the same location (within $\sim$50 m) where borehole temperatures of $-5.04 \pm 0.5°$C and $-5.50 \pm 0.5°$C were recorded at 14 m and 20 m depth respectively in May 2016 (Fig. 11). The $1.67°$C increase in temperature at 14 m depth over those 7 years supports the notion of a warming regional firn pack suggested by changes in borehole temperatures and stratigraphy at the Kaskawulsh/Hubbard Divide between 1965 and 2018, where an increase in melt and refreeze was indicated by an increase in both the quantity and 320 thickness of ice layers and lenses observed in the firn (Ochwat et al., 2021; Grew and Mellor, 1966). The warming firn pack and increased melt are likely due to atmospheric warming over that period, which has been shown to be amplified at high elevations; downscaled NARR gridded surface air temperatures in the St. Elias show a 1979-2016 warming rate of $0.028°$Ca$^{-1}$ between



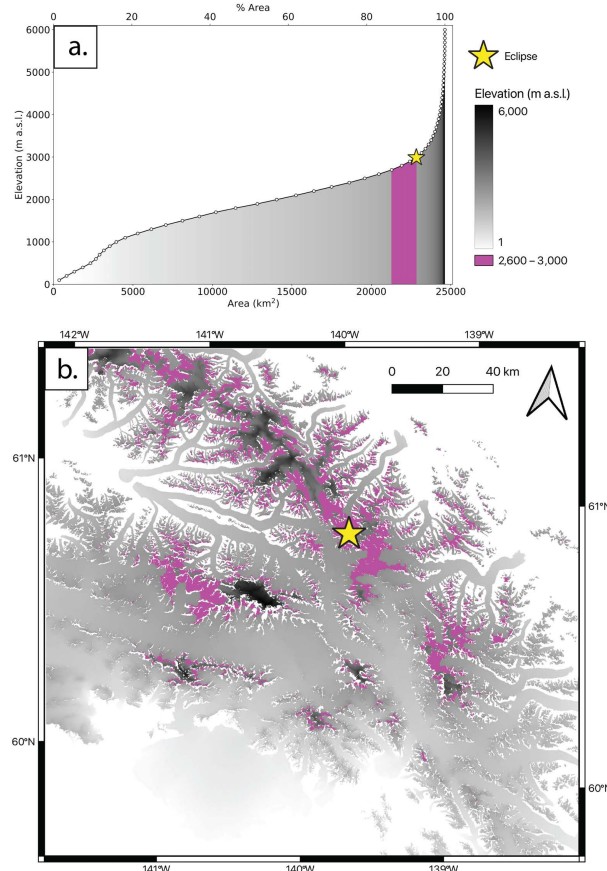

**Figure 11.** Hypsometric curve (a) and elevation map (b) of Eclipse Icefield and the surrounding area. Elevation is shown by the grayscale shading, with higher elevations indicated by darker grays. Elevations between 2,600 ma.s.l. and 3,000 ma.s.l. (the approximate elevations of the Kaskawulsh/Hubbard divide and Eclipse) are shown in magenta. The location of Eclipse is indicated with a yellow star.

5,500 and 6,000 ma.s.l., approximately 1.6 times larger than the global-average warming rate from 1979-2015 (Williamson et al., 2020).

Additionally, Eclipse density profiles from 2002, 2016, 2017 and 2023 show an apparent densification of the top 20 m of firn at Eclipse over the 21-year period (Fig. 12). We interpret this apparent densification with caution because of the unrealistically high measured densities of ice layers in the 2023 core (> 917 kgm$^{-3}$); however, both peaks and cyclic variations in measured density are consistent with our stratigraphic observations, coinciding with ice layers and seasonal changes in firn grain size. Results suggest an increase in melt within the snow and firn, which leads to densification first by rounding snow grains, allowing
them to pack more closely, and eventually by filling in pore space and refreezing (Cuffey and Paterson, 2010; Sommerfeld et al., 1970).



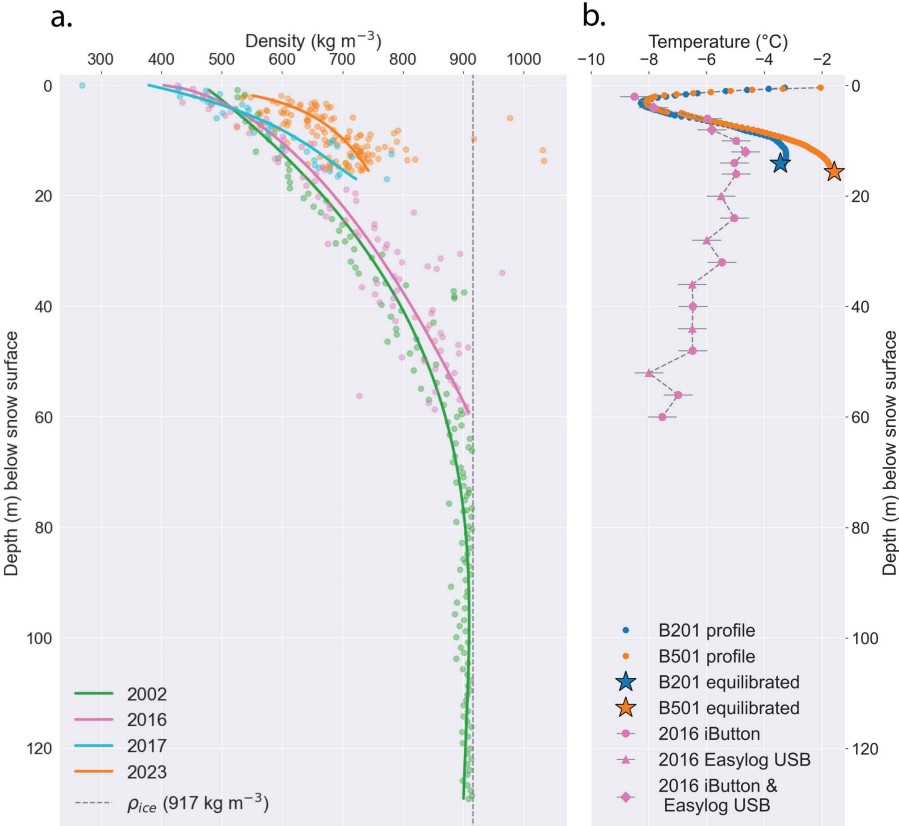

**Figure 12.** Eclipse 2023 (a) firn densities and (b) borehole temperatures in comparison to measurements at Eclipse from 2002, 2016 and 2017. Measured density values are shown by circles in panel (a) with a curve fitted to each year's data. The density of solid ice is shown in panel (a) by the dotted gray line. Curves are not fitted to temperature data in panel (b). Uncertainty in 2016 temperature measurements is shown by the gray error bars. Error bars for 2023 measurements are too short to be visible behind the markers.

## 4 Discussion

### 4.1 Meltwater movement and refreezing in snow

We see no evidence of seasonal melt onset in the top ∼4 m of snow. However, below this depth all three firn cores display a
variety of features associated with snowmelt, including ice layers, ice lenses, and bubbly and coarse-grained firn (Fig. 3). The presence of such features indicates that snow at Eclipse experiences melt and percolation during the summer months, leading to the formation of ice lenses and layers that freeze either in subsurface firn below 0°C or as ambient surface temperatures drop below freezing at night and during the fall and winter. Changes in surface snow structure in response to seasonal surface temperature changes are then preserved in the firn column as the surface is buried (Keegan et al., 2019).





In the early stages of summer melt, individual snow crystals undergo metamorphism that leads to rounding (Sommerfeld et al., 1970). Initially, capillary action causes the small amount of liquid water to stay in place between the crystals (Sommerfeld et al., 1970); if refreezing occurs at this stage, the refrozen melt only partially fills the boundaries between individual grains. It therefore does not produce a solid blue ice layer, instead appearing bubbly or composed of very coarse but well-sintered grains, what we note in our observations as "coarse-grained". The strength of these layers differentiates them from depth hoar and is

consistent with formation from meltwater, as repeated melt-freeze cycles increase both the size of snow crystals and the bond strength between them (Fierz et al., 2009; Sommerfeld et al., 1970).

    If the temperature remains above freezing for an extended period, individual snow crystals will continue to melt and vertical percolation will occur when the gravitational force acting on the liquid water exceeds the capillary forces holding it in the intergranular pore space (Colbeck, 1972). Capillary forces are inversely related to grain size, so fine-over-coarse grain transitions

between layers create capillary barriers that interrupt the vertical flow of water and allow it to move horizontally along the layer boundary (Jordan, 1995). Water will build up at the fine-over-coarse boundary until it is forced horizontally into pores the same size and tension of the smallest ones in the underlying layer, whereupon it can break through and re-establish vertical flow fingers below the transition (Jordan, 1995). Once ponding has occurred, wet grain growth is enhanced over the fine-over-coarse transition and the grain size transition eventually disappears, removing the capillary barrier to downward percolation

(McDowell et al., 2023). Although not necessary for ponding or grain growth, blue ice layers, which we observe in all three firn cores, may form from the saturated snow along fine-over-coarse transitions (McDowell et al., 2023). Lateral meltwater transport and refreezing may also occur along other impermeable barriers such as existing ice layers.

    Ice lenses (ice layers that do not extend across the entire core diameter) are observed in all three firn cores, indicating the development of preferential pathways for meltwater movement. As described above, percolation is a notoriously spatially

variable phenomenon, often occurring via vertical pipes that develop as surface melt progresses and can leave surrounding firn unaltered (Bengtsson, 1982). Preferential flow through vertical pipes has been observed extensively in both seasonal snowpacks (Williamson et al., 2020; Evans et al., 2016; Albert et al., 1999; Marsh and Woo, 1984) and glacier and ice sheet snow cover (Winski et al., 2012; Humphrey et al., 2012; Mernild et al., 2006; Bøggild et al., 2005). We interpret the appearance of ice lenses as evidence of vertical piping in the snow and firn at Eclipse although no pipes were sampled directly.

We interpret thin ($\leq 2\,\mathrm{mm}$) ice layers in all three cores as buried sun crusts. Sun crusts form when meltwater in the surface snow refreezes due to radiative cooling; the forming crust reduces shortwave absorption, allowing additional water vapor to condense below the initial glaze (Fierz et al., 2009). Buried sun crusts account for 31 % and 42 % of observed ice layers in cores B501 and B502 respectively, but only 16 % of observed ice layers in core B201. Sun crusts are surface formations that are later buried; they do not indicate movement of liquid melt through the snow and firn pack.

Thicker ice layers can form at the surface or deeper in the snow and firn, either through prolonged or intense individual melt events or through the cumulative effect of multiple melt events once an impermeable barrier to deeper percolation is established (Culberg et al., 2021). Multi-meter thick ice slabs, such as those observed in Greenland, can develop with multi-year meltwater production (Culberg et al., 2021). Alternatively, rapid freeze-thaw cycles can produce melt but inhibit its percolation, resulting in melt complexes that comprise many thin melt layers in close proximity, rather than thick consolidated slabs (Culberg et al.,




2021). Our results are more consistent with the latter scenario, with the thickest ice layers observed in cores B501 and B502 being 12.0 cm and 10.0 cm, respectively. However, the possibility of an ice slab at the bottom of core B201 cannot be dismissed, as we drilled through 33.0 cm of ice before stopping due to mechanical issues; the total thickness of that layer therefore remains unknown.

Neither our VHF (900 MHz) or HF (5 MHz) radar are high enough frequency to identify thin and discontinuous ice layers in the near-surface, which would indicate subsurface meltwater movement. However, both systems do provide a sense of the overall wetness of the firn pack. Liquid water is very effective at attenuating a radar signal, so the transition from a dry to wet firn pack can be inferred from the disappearance of stratigraphy in GPR data (e.g. Campbell et al., 2012). The presence of clear stratigraphy in both our VHF and HF radar profiles supports previous observations of an overall dry firn pack at Eclipse (McConnell, 2019). This is in contrast to a bright reflector atop washed out deeper stratigraphy seen in a 2017 radar profile from nearby Icefield Camp (∼400 m lower than Eclipse), indicative of a liquid water table in the firn (McConnell, 2019).

### 4.2 Melt percolation and spatial variability in firn character

GPR results, core stratigraphy, and borehole temperatures together indicate that while firn stratigraphy at Eclipse appears generally consistent on the kilometer scale, the effects of meltwater production and percolation across this distance can vary substantially. Because of the overall stratigraphic consistency down to at least ∼150 m (see Katmai layer labeled in Fig. 6c) across our radar profiles, we interpret differences among our firn cores' stratigraphy to result from uncertainty in both our GPR and firn core depth scales and heterogeneous meltwater percolation, rather than bulk snow deposition and firn densification processes.

Stratigraphic profiles from cores B501 and B502, drilled 0.6 m from each other, show that firn stratigraphy varies horizontally on sub-meter scales, but spatially consistent annual melt/freeze cycles can still be observed. The location, thickness, and type (e.g. clustered ice layers vs. coarse-grained firn) of melt features differ between the two cores due to small spatial variations in the amount of snow deposited at each site, and to the irregular nature of ice lens, finger, and layer formation. However, the cores do share certain macroscale features. For example, both cores show evidence of meltwater alteration throughout the ∼2 m below the LSS, a thick (7–10 cm) ice layer between 1.4 and 1.8 m BLSS, fine-grained firn between ∼2 and 3 m BLSS (which we interpret to be winter accumulation), and a section of coarse-grained firn between ∼3 and 4 m BLSS. Annual cycles as approximated by transitions between ice-rich and unaltered sections of firn are consistent with those suggested by cyclic variations in firn density. Ice-rich core sections correspond to regions of low, rather than high, density because the background firn is composed of summer snowfall, which tends to have larger crystals and pack down into a less dense firn layer compared to snowfall under colder winter conditions (Albert and Shultz, 2002).

Core B201, drilled ∼1.5 km from the B5 cores and at a similar elevation, shows some similarity with them in macroscale features such as the location of winter accumulation layers, indicated by fine-grained ice-free firn. However, B201 has a higher firn ice content (10 % by volume) than either B501 (5 % by volume) or B502 (3 % by volume), evidence of either greater surface melt production or greater retention of liquid water in the near-surface. Additionally, B201's ice content tends to comprise thicker ice layers despite B201 having lower borehole temperatures than B501 below ∼10 m depth. In core B201,



48 % of observed ice layers were > 1 cm thick, accounting for 93 % of the core's total ice content, while 28 % and 11 % of
observed ice layers in cores B501 and B502 were > 1 cm thick, accounting for 82 % and 71 % of each core's total ice content.
Both B201's higher ice content and tendency toward thicker ice layers hold when compared with B501 over only the 14.15
m in which the two cores overlap, ensuring that both cores contain the same proportions of summer and winter snowfall over
their length.

As mentioned above, B501 and B201 borehole temperatures indicate that firn at 14 m depth can vary in temperature by
>1.5°C within two kilometers. This difference in temperature is likely due to a difference in the surface energy balance and/or
meltwater input at the two sites rather than a difference in atmospheric temperature. Fourteen meters is below the typical
penetration of the annual temperature wave via conduction (Cuffey and Paterson, 2010); firn at this depth is insulated from
surface conditions by the overlying firn and snow and typically reflects mean annual temperature, which is virtually the same at
our two study sites. However, the relative magnitude of surface energy balance terms can vary substantially based on prevailing
weather, topographic shading and seasonal effects, even between sites with similar mean annual temperatures (Hannah et al.,
2000). One explanation for observed differences between sites B2 and B5 is that B5 on average receives slightly more solar
radiation due to its southeast aspect, especially in the winter. If this effect is predominantly a winter phenomenon, this could
explain why B5 has fewer melt features despite being slightly warmer. However, in areas with surface melt and percolation
into the subsurface, the role of conduction in downward heat transport is comparatively minor relative to that of latent heat
associated with the refreezing of meltwater (Cuffey and Paterson, 2010). Another explanation is that meltwater percolation and
refreezing is responsible for warming the firn at B5, but occurs adjacent to our core sites and therefore is not recorded in our
core stratigraphy. For example, the lateral transport of liquid water along a subsurface layer boundary from the southeast-facing
areas upslope of site B5 may account for the higher subsurface heat content of the site relative to B2. However, subsurface
meltwater flow would be limited by the large cold content of below-freezing firn. Additionally, site B2 could also experience
liquid water transport from upslope areas, predominantly with a western aspect; though based on surface debris, the areas
upslope of B2 experience greater avalanche disruption than those upslope of B5, making meltwater transport along consistent
subsurface pathways less likely.

### 4.3 Development of temperate firn

The present difference in firn water content between Eclipse, Icefield Camp and Kaskawulsh is likely because Eclipse sits
~400 m higher in elevation than the other sites. The lower elevation, but generally similar environment, of Icefield Camp and
Kaskawulsh to Eclipse make them useful case studies for predicting the future firn evolution at Eclipse with continued warming
in the St. Elias Range.

Unlike the Kaskawulsh/Hubbard Divide area, Eclipse has yet to develop a firn aquifer, as demonstrated by GPR data from
both 2016 and 2023. HF (5-10 MHz) data from both years show bedrock that slopes to a trough in the middle of the icefield,
a bright reflector at ~150 m depth interpreted as the ash layer from the 1912 Katmai eruption, and continuous stratigraphy
with the exception of surface zones of avalanche debris. Neither 2016 nor 2023 HF data show a bright reflector indicative of a
liquid water table. VHF (400–900 MHz) data from both years also look similar, with clear and continuous stratigraphy down





to 20 m depth indicative of a dry firn pack. Despite remaining dry to date, borehole temperatures indicate that at least part of Eclipse Icefield is < 2°C from supporting liquid water at depth. Moreover, model results indicate that Eclipse is close to the threshold for developing temperate firn, with a 2 % chance of year-round temperate firn at 15 m depth by 2033 without continued atmospheric warming, a 51 % chance with 0.5°C warming, and a 98 % chance with 1°C warming. Model behavior also suggests an increase in liquid water content at depth, with the potential for firn aquifer development. During many of our model runs, the CFM failed to produce any outputs below ~25 m depth starting in the late 2020s. Because the CFM is limited in its ability to handle large amounts of liquid water, we associate this pattern of model failure with high amounts of persistent liquid water, consistent with the possibility that Eclipse follows a similar trajectory to Kaskawulsh and Icefield Camp, developing a firn aquifer in the next 5–10 years.

We suggest that increased extreme melt events during the height of the melt season promote the development of year-round temperate firn in the St. Elias. Model results for Eclipse show the development of year-round temperate firn at 15 m depth associated with an increase in total PDDs throughout the melt season and more extreme individual melt events, rather than a greater number of melt events or prolonged melt season (Fig. 10, Tables B1-B2). In Greenland, extreme melt events have been related to firn's multi-year response to surface melt via the formation of thick ice slabs and ice layer complexes, which cause a near-surface barrier to downward percolation (Culberg et al., 2021). In the St. Elias, however, extreme melt events are more likely to result in sustained heat transport to depth because of the insulating effect of the region's high annual accumulation (1.4 mw.e.a$^{-1}$ at Eclipse) relative to accumulation rates in Greenland (0.3 – 1.2 mw.e.a$^{-1}$; Hawley et al., 2020; Montgomery et al., 2020; Burgess et al., 2010).

If Eclipse does indeed develop a firn aquifer in the next decade, over 90 % (>22,000 km$^2$) of the region from 59.59°N to 61.58°N and 138.02°W to 142.23°W could support liquid water in the firn based on its elevation (Fig. 11). Moreover, areas above Eclipse in elevation are largely steep mountain peaks and represent a far smaller portion of the St. Elias hydrological reservoir than the broad icefields below, which retain more snow and ice. Deep (> 10 m depth) temperate firn up to 3,000 ma.s.l. would therefore represent widespread meltwater percolation and constitute a wholesale change in the region's hydrological system. In particular, the capacity of the icefields to buffer runoff would be reduced by meltwater's direct occupation of pore space and by intensified compaction associated with its rounding and lubrication of grains (Thompson-Munson et al., 2024; Amory et al., 2024; Colbeck, 1982; Colbeck and Parssinen, 1978). Such processes associated with warming have contributed to a 5 % reduction in firn pore space in Greenland since 1980 (Amory et al., 2024). Additionally, the loss of firn pore space in response to atmospheric warming is amplified relative to its gain in response to atmospheric cooling, meaning observed densification of the firn to date has long-term consequences for runoff buffering (Thompson-Munson et al., 2024).

## 4.4 Melt percolation and ice coring

Prior work at Eclipse has included the recovery of three deep ice cores: 160 m in 1996, and 345 m and 130 m in 2002 (Yalcin et al., 2006; Wake et al., 2002). More recently, shallow (59 m and 17 m) firn cores were drilled at the site in 2016 and 2017 (McConnell, 2019). Because of its high elevation (3,017 ma.s.l.), high accumulation rate (1.4 mw.e.a$^{-1}$) and thick ice (> 650 m), Eclipse Icefield presents a unique opportunity for the recovery of a relatively long, high-resolution climate record from the





Yukon/Alaska region (McConnell, 2019). However, recent and continued warming threaten the viability of coring efforts and the preservation of a climate record at Eclipse as percolating melt can influence paleoproxy records by homogenizing isotope and chemical signals in the snow and firn through which it travels (Moran and Marshall, 2009). In addition, liquid water in the

480 snow and firn pack can limit or preclude ice core recovery with mechanical ice drills, even if the climate record remains stable.

Results from our firn core stratigraphy, borehole temperatures, and GPR surveys suggest that although there is some melt-water movement through the snow and firn at Eclipse, it is not enough to characterize Eclipse as a "wet" site or exclude it from ice coring. Firn core stratigraphy shows numerous melt layers, but the presence of melt layers alone is not enough to preclude climate interpretation; climate records have successfully been developed from ice cores recovered at other sites with melt lay-

485 ers. In fact, when found in otherwise cold, dry and unaltered firn, melt layers can be used to recreate surface temperatures at the time of their formation (Winski et al., 2018). Eclipse may therefore currently be an ideal site for recovering and developing a melt layer record of past regional climate. Borehole temperatures also support the suggestion that the Eclipse firn pack remains cold enough for ice core recovery, remaining below the melting point with a maximum firn temperature of –1.62 ± 0.01°C. Finally, both our VHF (900 MHz) and HF (5 MHz) radar profiles show clearly visible stratigraphy indicative of a firn pack with

490 limited liquid water, and neither contains any horizon resembling a liquid water table in the firn. All of this is consistent with isotope data from 2016 showing clear annual oscillations, with no apparent dampening by movement of meltwater through the firn (McConnell, 2019).

Because the firn at Eclipse is close to temperate in places, the heterogeneity of meltwater movement and its effects on the firn must be taken into account for the successful recovery of ice core climate records from the site. Specifically, we suggest

a drilling campaign is more likely to be successful near site B2 as opposed to site B5, especially with continued warming and melt production. Moreover, opportunities for ice coring in the St. Elias more broadly may be severely limited with continued atmospheric warming and the associated development of temperate firn. Ice coring efforts in the region should therefore be undertaken as soon as possible to maximize the chance of success.

## 5 Conclusions

Stratigraphy, density, and borehole temperature data from Eclipse Icefield indicate that although the site can still be charac-terized as largely "dry", the production and percolation of meltwater is present and increasing. Numerous ice layers, lenses, and regions of melt-affected firn are seen in all three firn cores recovered in 2023, with stratigraphic observations in core B501 supported by density data. Borehole temperatures indicate that from 2016 to 2023 there has been a 1.67°C warming of the firn at 14 m depth, and model results indicate that warming of the firn below 10 m depth may continue over the next decade, with

a 2 % chance of becoming temperate year-round at 15 m depth by 2033, even without continued atmospheric warming. The chance of developing year-round temperate firn at 15 m depth remains around 2 % with 0.1°C atmospheric warming by 2033, but increases to 12 % with 0.2°C warming, 51 % with 0.5°C warming and 98 % with 1°C warming over the same period. Development of year-round temperate firn at Eclipse is associated with an increase in total PDDs throughout the melt season




and more extreme individual melt events rather than a greater number of melt events or prolonged melt season. Extreme melt
events combined with the site's high ($1.4 \ \mathrm{mw.e.a}^{-1}$) annual accumulation likely result in sustained heat transport to depth.

As > 90 % of the region from 59.59°N to 61.58°N and 138.02°W to 142.23°W is below Eclipse in elevation, the development
of temperate firn and/or a firn aquifer at Eclipse would represent the ability for widespread meltwater runoff across the St. Elias
Range, indicating a wholesale change in the region's hydrological system and a reduction in its ability to buffer runoff. Given its
relatively dry conditions to date, Eclipse remains a site of interest for recovering a long-term regional climate record. However,
because the firn at Eclipse is close to temperate in places, the heterogeneity of meltwater movement and its effects on the firn
must be taken into account for the successful recovery of such a record. Moreover, opportunities for ice coring may be limited
with continued atmospheric warming and the associated development of temperate firn and/or a firn aquifer. Such a change in
firn character at Eclipse and across areas at or below similar elevations would severely limit potential ice core sites in the St.
Elias.

*Code availability.* Code for the Community Firn Model an be downloaded from Github at https://github.com/UWGlaciology/CommunityFirnModel

## Appendix A: Community Firn Model sensitivity tests

We test the sensitivity of the Community Firn Model to the air temperatures used during the model spinup by using four
different spinup schemes based on different air temperature data. First, we use 2013 to 2024 in situ data from an automatic
weather station (AWS) near the Kaskawulsh/Hubbard ice divide ("Divide AWS", Fig. 1), which we repeat for the duration
of the spinup. Second, we generate synthetic climate data by randomly selecting daily temperature values from a Gaussian
distribution described by the mean and standard deviation of the Divide AWS data for each day of the year, which have been
corrected for the difference in elevation between Divide and Eclipse using a lapse rate of $-3.98°\mathrm{C \ km}^{-1}$ (Hill et al., 2021). Our
third spinup scheme uses the same Gaussian method, but we apply a temperature correction to account for a historical warming
rate of $0.024°\mathrm{Ca}^{-1}$ between 1979 and 2016 (Williamson et al., 2020). Finally, we use downscaled NARR temperature data
from 1983 to 2013 (Jarosch et al., 2012). We use the same mean annual accumulation rate ($1.4 \ \mathrm{mw.e.a}^{-1}$) under all spinup
schemes.

The model is always forced with the same Divide AWS air temperatures from 2013–2024 regardless of spinup regime, so
surface temperatures (0 m depth) from 2013 to 2024 are consistent among all four spinup schemes (Fig. A1) and there is
no significant difference among the four spinup schemes at 0 m depth (Kruskal-Wallis test; $H = 0.51$, $p \geq 0.05$). There are,
however, significant differences among the four schemes (Kruskal-Wallis test) at 5 m ($H = 65.81$, $p < 0.05$), 10 m ($H = 485.92$,
$p < 0.05$), 15 m ($H = 1535.42$, $p < 0.05$), and 20 m ($H = 4008.88$, $p < 0.05$) depth. We select downscaled NARR air temperatures
as the forcing data for our reference model spinup because they provide the most conservative baseline firn temperatures below
10 m depth (Fig. A1). We therefore take all predictions of firn warming to be conservative estimates.





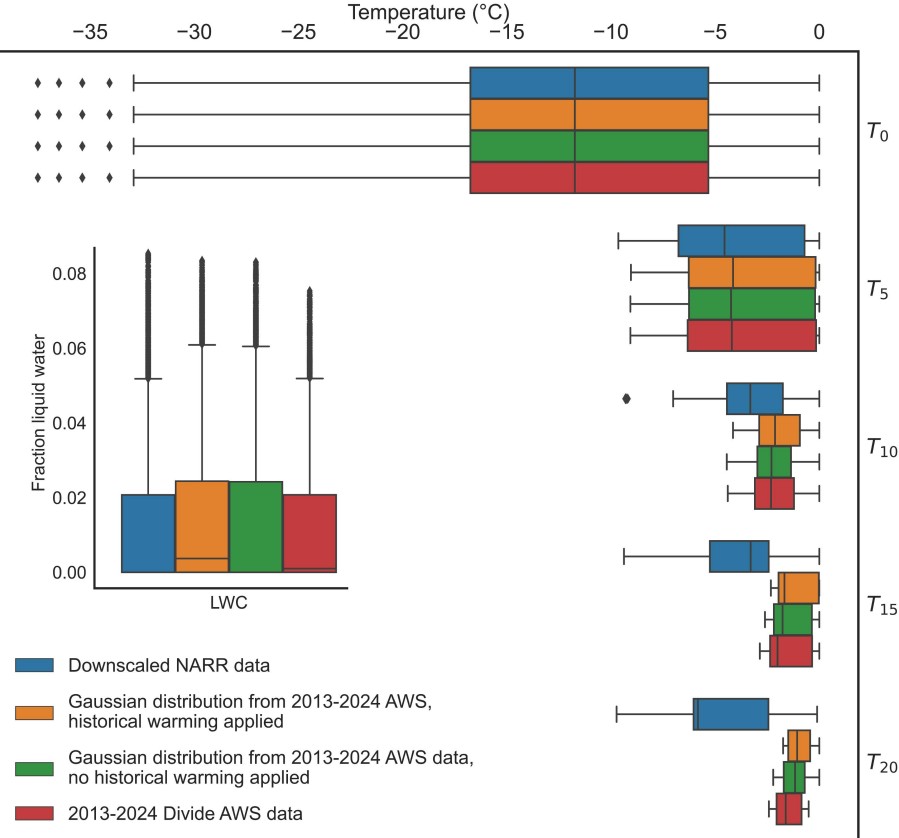

**Figure A1.** Temperature and liquid water content (LWC) sensitivity to spinup scheme. Distributions include temperature and LWC outputs from 2013–2024 under four different spinup scenarios. Distributions are shown for temperature at 0 m, 5 m, 10 m, 15 m, and 20 m depth. LWC is calculated for the entire firn column.

We also test the model sensitivity to the degree day factor (DDF) and surface density values used in surface melt production
and firn densification. Across the three hundred combinations of DDF and surface density that we tested, the CFM produced firn temperatures ranging from 0°C to nearly –10°C in May/June 2023, when borehole temperatures indicated firn at ~14 m depth at Eclipse to be between –2°C and –4°C (Fig. A2). Firn temperatures at 0 m, 5 m, 10 m, 15 m, and 20 m depth, as well as total firn column liquid water content, are shown for all sensitivity test runs that produced 2023 firn temperatures consistent with borehole measurements in Figure A3. Results of our sensitivity tests for model runs spun up with downscaled NARR air
temperatures are shown in Figure A4 with our selected reference model (used for all reported results) indicated.



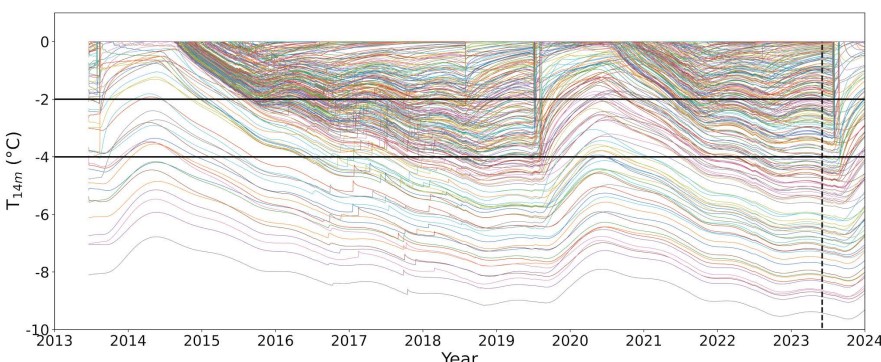

**Figure A2.** Evolution of firn temperature at 14 m depth under 300 combinations of degree day factor (DDF) and surface density. Dotted vertical line marks the date of in situ borehole temperature measurements in May/June 2023. Solid horizontal lines at –2°C and –4°C show the bounds of realistic May/June 2023 firn temperatures at 14 m depth based on borehole measurements.





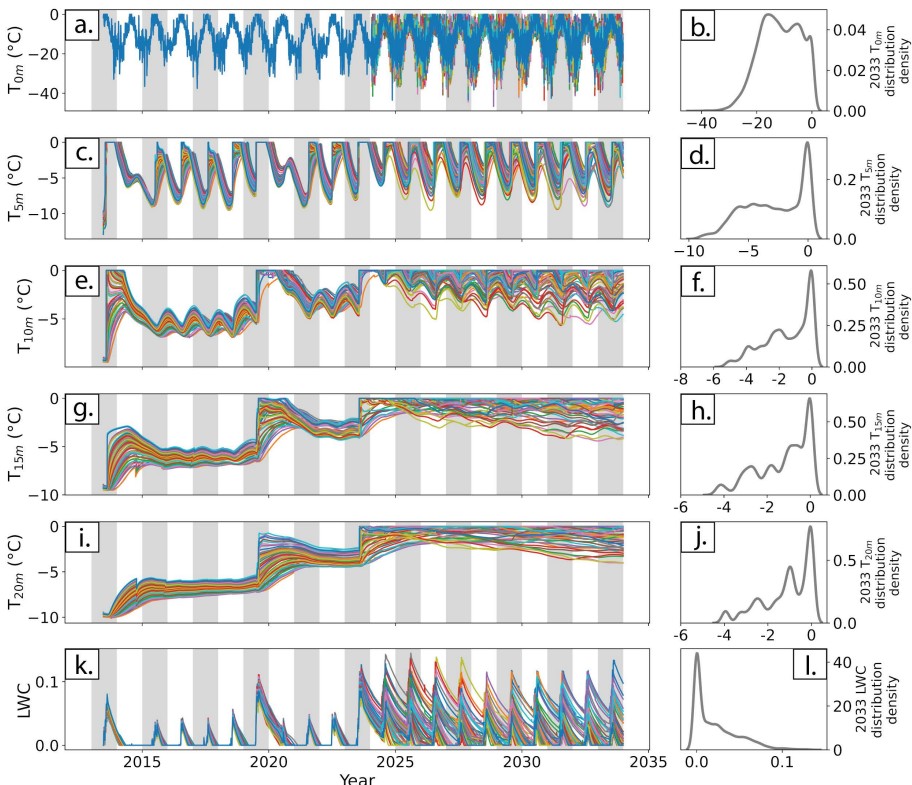

**Figure A3.** Temperature (a-j) and liquid water content (LWC; k-l) timeseries and distribution plots for model runs that produce realistic results at Eclipse Icefield. Temperatures are shown at 0 m (a-b), 5 m (c-d), 10 m (e-f), 15 m (g-h), 20 m (i-j) depth. LWC is calculated for the entire firn column. All model runs were forced using downscaled NARR air temperatures during the spinp period (pre-2013), elevation-corrected Divide AWS data from 2013–2024, and temperatures randomly drawn from a Gaussian distribution described by the mean and standard deviation of elevation-corrected 2013–2024 Divide AWS temperatures from 2024–2033. Forcing temperatures from 2024–2033 assume a continuation of current climate conditions. Gray bars show odd-numbered years (e.g. 2015, 2017, etc.). Values on the x-axis of the righthand panels (distributions) correspond to those on the y-axis of the lefthand panels (timeseries). Distributions are shown for 2033 outputs only.





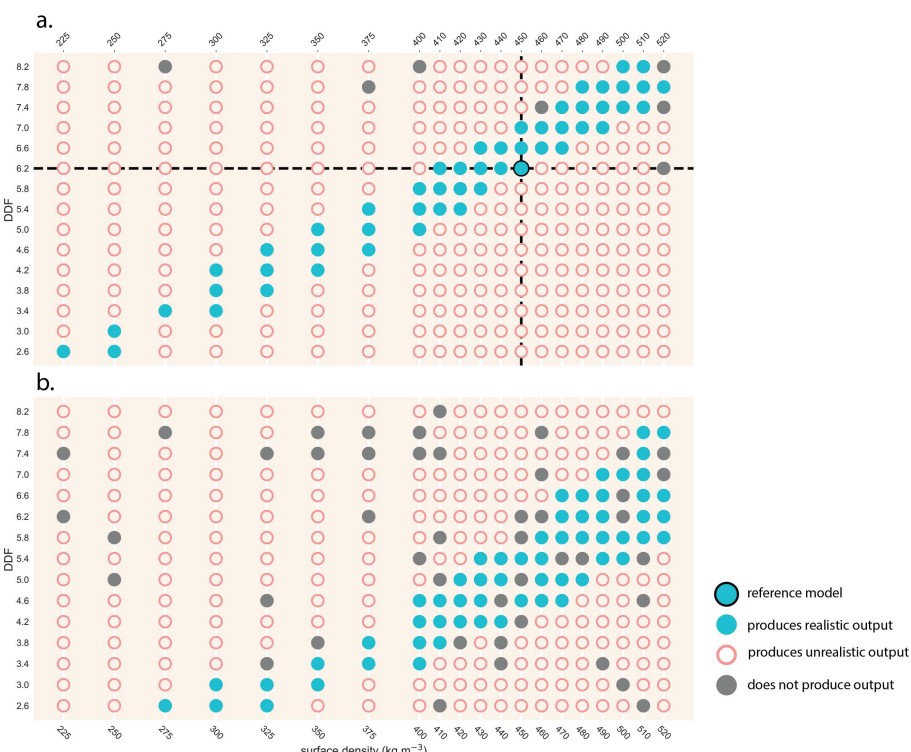

**Figure A4.** Degree Day Factor (DDF) and surface density pairings that produce realistic (blue filled circles), unrealistic (pink open circles), and no (gray filled circles) outputs at Eclipse. Panel (a) shows model runs spun up with downscaled NARR temperatures. Panel (b) shows model runs spun up with temperature values randomly selected from a Gaussian distribution based on elevation-corrected Divide AWS data with a historical warming rate of $0.024°\mathrm{Ca}^{-1}$ between 1979 and 2016 applied (Williamson et al., 2020). Our reference model is indicated by the black dashed lines and black-outlined circle.





## Appendix B: Community Firn Model runs that produce temperate firn

**Table B1.** Median positive degree days (PDDs) over May through September and Wilcoxon Rank Sum results for model runs that develop year-round temperate firn by 2033 vs. those that do not.

| Year | Median PDDs (temperate, °C) | Median PDDs (polythermal, °C) | W | p |
|------|------|------|------|------|
| 2024 | 36.68 | 34.08 | 1.66 | $\geq 0.05$ |
| 2025 | 37.20 | 32.34 | 2.20 | $< 0.05$ |
| 2026 | 37.81 | 33.87 | 2.61 | $< 0.05$ |
| 2027 | 39.57 | 33.85 | 3.17 | $< 0.05$ |
| 2028 | 41.86 | 32.06 | 6.26 | $< 0.05$ |
| 2029 | 42.10 | 32.03 | 6.67 | $< 0.05$ |
| 2030 | 50.93 | 33.46 | 7.91 | $< 0.05$ |
| 2031 | 49.12 | 33.80 | 8.13 | $< 0.05$ |
| 2032 | 54.19 | 33.31 | 9.29 | $< 0.05$ |
| 2033 | 55.24 | 35.56 | 9.04 | $< 0.05$ |





**Table B2.** Results of Wilcoxon Rank Sum tests between model runs that produce year-round temperate firn by 2033 and those that do not.

| Climate scenario | Number of melt events | | Melt event magnitude | | Melt season start | | Melt season end | | Melt season length | |
|---|---|---|---|---|---|---|---|---|---|---|
| | W | p | W | p | W | p | W | p | W | p |
| −0.1°C | n/a | n/a | n/a | n/a | n/a | n/a | n/a | n/a | n/a | n/a |
| Current climate conditions | -1.07 | ≥ 0.05 | 1.72 | ≥ 0.05 | 2.03 | ≥ 0.05 | -1.23 | < 0.05 | -2.54 | ≥ 0.05 |
| +0.01°C | 1.11 | ≥ 0.05 | 1.37 | ≥ 0.05 | -1.13 | < 0.05 | -0.49 | < 0.05 | 0.60 | < 0.05 |
| +0.02°C | -1.09 | ≥ 0.05 | 2.53 | < 0.05 | 0.67 | < 0.05 | -0.41 | < 0.05 | -0.75 | < 0.05 |
| +0.05°C | 1.13 | ≥ 0.05 | 2.41 | < 0.05 | 0.13 | < 0.05 | 0.14 | < 0.05 | -0.21 | < 0.05 |
| +0.1°C | 0.95 | ≥ 0.05 | 0.37 | ≥ 0.05 | -0.10 | < 0.05 | 2.05 | < 0.05 | 1.17 | < 0.05 |



*Author contributions.* IK, DW and KK formulated the research goals, hypotheses and testing methods. IK, ES, MM, RC, and JH participated in fieldwork and data collection. IK completed data analysis with contributions from ES, CMS, MA and SW. IK prepared the manuscript with contributions from all co-authors

*Competing interests.* The contact author has declared that none of the authors has any competing interests.

*Acknowledgements.* We thank the U.S. National Science Foundation, Golden Family Foundation, Bob and Judy Sturgis Exploration Fund, Natural Sciences and Engineering Research Council of Canada, Polar Continental Shelf Program, American Alpine Club, American Geophysical Union, Geophysical Survey Systems Inc., and the University of Maine Graduate Student Government for supporting work in the St. Elias Range. We also thank Stefan Bastien, Steven Bernsen, Erik Blake, Seth Campbell, Dan Dixon, William Kochtitzky, Justin Leavitt, Brit-

tany Main, Dorota Medrzycka, Alex Mondrick, Patrick Saylor, and Cameron Wake for their efforts in obtaining field data. Finally, we thank Kluane National Park and Reserve, the Kluane and Champagne and Aishihik First Nations, Icefield Discovery, and Kluane Lake Research Station for collaboration and support.



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
