# Peer review of "Ongoing firn warming at Eclipse Icefield, Yukon, indicates potential widespread meltwater percolation and retention in firn pack across the St. Elias Range"

_EGUsphere, 2024_

## Author Comment (AC1)

**Reviewer 1**

**General comments**
**General comment 1**

In the paper you state "Model results for Eclipse show the development of year-round temperate firn at 15 m depth associated with an increase in total PDDs throughout the melt season and more extreme individual melt events, rather than a greater number of melt events or prolonged melt season" (Lines 453 – 455). This statement seems to support findings of others such as Nghiem et al. (2012) and Horlings et al. (2022) that show summers of extreme melting in Greenland are mostly dominated by intense but short-lived events. I was wondering if you could dive into this some more? You show the number of PDDs for temperate vs polythermal firn across all model runs. Are there other ways to look at this that might be as/more useful to justify this conclusion? What about calculating the average PDD (or meltwater input) per melt event for temperate vs. polythermal firn? In my mind that should show the intensification of melt events that produce temperate firn.

However, I'm confused with this conclusion on lines 453 – 455 because in Table B2 you report p values less than 0.05 indicating significant differences in the melt season start, end, and length between model runs that produce temperate firn and those that do not with 0.2 and 0.5 °C of warming, so how do you rule out that prolonged melt seasons are not driving the creation of a temperate firn layer?

Additionally, in combination with intense melt events causing firn aquifer expansion, Horlings et al. (2022) found that winter temperatures substantially increased and the firn's cold content significantly decreased. Do you see similar trends? For example, do you find that when the firn becomes temperate there is less of a regeneration of cold content in the winter? Or is it primarily a summer-driven process?

I think fleshing out some of the drivers of firn warming/possible aquifer formation would be really interesting and useful for the firn community.

> *Thank you for such a careful read of the appendix. To summarize our responses to your line comments below:*
> - *We've updated the in-text significance values in parentheses to be less confusing*
> - *The p-values in Table B2 appear to have fallen prey to sloppy copy/pasting - Good catch, we're sorry about that!*
> - *We've rerun all the stats after using a seasonally variable accumulation rate, and updated the results, discussion and appendix accordingly*
>
> *In rerunning all the values for Table B2, we've adjusted our comparisons to be by year, rather than by climate scenario, to provide a more meaningful analysis of the patterns of melt events over a series of years that lead to temperate vs. polythermal firn - this incorporates your suggestion to calculate PDDs per melt event.*

*We have replaced problematic text that you identified in your comments with the following:*

**Methods:**
*To examine the conditions associated with the production of year-round temperate firn at 15 m depth, we quantify the mean winter temperature, melt season start, melt season end, melt season length and total PDDs each year. We also quantify the number of individual melt events and the magnitude of each event. We define melt events as any period over which the snow surface is continuously melting without refreezing.*

**Results:**
*Conditions associated with the development of year-round temperate firn at 15 m depth include lower mean winter temperature, higher total melt season PDDs and greater magnitude of individual melt events. We find a significant (p < 0.05) difference (independent samples t-test) for mean winter temperatures during all years after 2025 between model runs that do produce year-round temperate firn by 2033 and those that do not (Table B1).*

*Additionally, for model runs that do not produce temperate firn, the median of the total melt season PDDs (over all model runs for any given year) never exceeds 35 (Fig. 10). In contrast, the median total melt season PDDs over all model runs that do produce temperate firn by 2033 ranges from 31.58 in 2025 to 52.88 in 2033. Finally, we find a significant (p < 0.05) difference (Wilcoxon Rank Sum test) for the total melt season PDDs, the number of individual melt events, and the median melt event magnitude (mm) between model runs that produce temperate firn at 15 m depth and those that don't during most years 2024-2033 (Table B2). We do not find any significant (p ≥ 0.05) difference in the number of melt events, nor in median melt season start, end, or length (Table B3). Complete results of our statistical tests are shown in Appendix B.*

**Discussion**
*We suggest that increased extreme melt events during the height of summer promote the development of year-round temperate firn in the St. Elias. Model results for Eclipse show the development of year-round temperate firn at 15 m depth associated with an increase in total PDDs throughout the melt season, as well as with a greater number and more extreme melt events, rather than an earlier or prolonged melt season (Fig. 10, Tables B2-B3). In Greenland, extreme melt events have been related to firn's multi-year response to surface melt via the formation of thick ice slabs and ice layer complexes, which cause a near-surface barrier to downward percolation (Culberg et al., 2021). In the St. Elias, however, extreme melt events are more likely to result in sustained heat transport to depth because of the insulating effect of the region's high annual accumulation (1.4 m w.e. a$^{-1}$ at Eclipse) relative to accumulation rates in Greenland (0.3 – 1.2 m w.e. a$^{-1}$; Hawley et al., 2020; Montgomery et al., 2020; Burgess et al., 2010.*

*Also consistent with observations in Greenland (Horlings et al., 2022), our results show that the development of year-round temperate firn at Eclipse is associated with an increase in winter temperatures. Rather than directly relating to melt production, wintertime warming affects firn properties by reducing the regeneration of cold content that occurs between melt seasons, effectively enabling the warming effects of summertime melt to compound from one melt season to the next.*

**Appendix B**

**Appendix B: Community Firn Model runs that produce temperate firn**

**Table B1.** Comparison of winter (October through March) mean temperatures between model runs that produce year-round temperate firn by 2033 and those that do not.

| Year | Temperate mean (°C) | Polythermal mean (°C) | T | p |
|------|------|------|------|------|
| 2024 | -16.63 | -16.66 | 0.77 | $p \geq 0.05$ |
| 2025 | -16.46 | -16.53 | 1.49 | $p \geq 0.05$ |
| 2026 | -16.40 | -15.54 | 2.84 | $p < 0.05$ |
| 2027 | -16.35 | -16.51 | 3.34 | $p < 0.05$ |
| 2028 | -16.27 | -16.55 | 5.89 | $p < 0.05$ |
| 2029 | -16.20 | -16.56 | 7.48 | $p < 0.05$ |
| 2030 | -16.08 | -16.50 | 8.87 | $p < 0.05$ |
| 2031 | -16.07 | -16.49 | 8.78 | $p < 0.05$ |
| 2032 | -16.01 | -16.50 | 10.23 | $p < 0.05$ |
| 2033 | -15.88 | -16.44 | 11.94 | $p < 0.05$ |

**Table B2.** Melt event number and magnitude results of Wilcoxon Rank Sum tests between model runs that produce year-round temperate firn by 2033 and those that do not.

| Year | Median PDDs | | Number of melt events | | Melt event magnitude | |
|------|------|------|------|------|------|------|
| | W | p | W | p | W | p |
| 2024 | -0.13 | $\geq 0.05$ | 0.02 | $p \geq 0.05$ | 0.27 | $p \geq 0.05$ |
| 2025 | -0.10 | $\geq 0.05$ | 0.75 | $p \geq 0.05$ | 0.00 | $p \geq 0.05$ |
| 2026 | 4.68 | $< 0.05$ | 1.23 | $p \geq 0.05$ | 2.74 | $p < 0.05$ |
| 2027 | 4.73 | $< 0.05$ | 2.87 | $p < 0.05$ | 3.03 | $p < 0.05$ |
| 2028 | 4.72 | $< 0.05$ | 2.82 | $p < 0.05$ | 1.17 | $p \geq 0.05$ |
| 2029 | 6.01 | $< 0.05$ | 3.88 | $p < 0.05$ | 3.06 | $p < 0.05$ |
| 2030 | 7.63 | $< 0.05$ | 4.22 | $p < 0.05$ | 3.81 | $p < 0.05$ |
| 2031 | 8.87 | $< 0.05$ | 5.19 | $p < 0.05$ | 5.47 | $p < 0.05$ |
| 2032 | 9.16 | $< 0.05$ | 6.17 | $p < 0.05$ | 5.45 | $p < 0.05$ |
| 2033 | 8.26 | $< 0.05$ | 6.70 | $p < 0.05$ | 3.21 | $p < 0.05$ |

**Table B3.** Melt season timing and duration results of Wilcoxon Rank Sum tests between model runs that produce year-round temperate firn by 2033 and those that do not.

| Year | Melt season start | | Melt season end | | Melt season length | |
|------|------|------|------|------|------|------|
| | W | p | W | p | W | p |
| 2024 | -0.18 | p ≥ 0.05 | 0.87 | p ≥ 0.05 | 0.87 | p ≥ 0.05 |
| 2025 | 0.24 | p ≥ 0.05 | 0.89 | p ≥ 0.05 | -0.15 | p ≥ 0.05 |
| 2026 | 1.27 | p ≥ 0.05 | -1.65 | p ≥ 0.05 | -1.66 | p ≥ 0.05 |
| 2027 | 1.36 | p ≥ 0.05 | -1.90 | p ≥ 0.05 | -1.72 | p ≥ 0.05 |
| 2028 | 0.90 | p ≥ 0.05 | 0.34 | p ≥ 0.05 | -0.44 | p ≥ 0.05 |
| 2029 | -0.02 | p ≥ 0.05 | -0.67 | p ≥ 0.05 | -1.32 | p ≥ 0.05 |
| 2030 | -0.21 | p ≥ 0.05 | 1.55 p | p ≥ 0.05 | 1.50 | p ≥ 0.05 |
| 2031 | 0.41 | p ≥ 0.05 | -1.15 | p ≥ 0.05 | -1.55 | p ≥ 0.05 |
| 2032 | 1.20 | p ≥ 0.05 | 1.06 | p ≥ 0.05 | -0.36 | p ≥ 0.05 |
| 2033 | -0.74 | p ≥ 0.05 | 1.08 | p ≥ 0.05 | 1.37 | p ≥ 0.05 |

**General comment 2**

This paper has a lot of components to it, all of which are interesting on their own, but can sometimes make the paper feel a bit disjointed. To me, it seems as though you are trying to make two main points with the paper: (1) that the deep firn column is warming from increased latent heat inputs, and there is a high likelihood that Eclipse will become temperate by 2033; and (2) although there are melt inputs into the firn, the firn column here is relatively "dry" and can still (for now) be used as an ice coring site. This may be my personal research bias, but it seems like point number 1 is the "highlight" takeaway, and it is the point which it appears you spend the most time analyzing with the modeling study. For me, I find there is a lot to keep track of and I would urge the authors to consider ways to make the paper more concise and punchier. Could you structure the discussion to just more concisely say (1) say the firn column is still relatively dry (from GPR and cores), so it could still be a valuable ice core site. (2) but there is evidence of increasing meltwater inputs (firn density, temperature increases), (3) firn modeling shows that there is a likelihood of the firn becoming temperate. I also think you could move the hypsometry to the Appendix and just reference it when mentioning that 90% of the Eclipse icefield might be at risk for firn aquifer development? I am just trying to make sure the key points aren't lost, but feel free to disagree with these suggestions.

*As someone who greatly appreciates a clear, concise takeaway message, we appreciate you pointing out the double-takeaway problem here! We've now eliminated section 4.4 (melt percolation and ice coring), and instead added a greatly shortened intro paragraph at the beginning of the discussion section based on your suggested framing.*

*Results from our firn core stratigraphy, borehole temperatures, and GPR surveys (discussed in greater detail below) suggest that although there is some meltwater movement through the snow and firn at Eclipse, it is not enough to characterize the site as "wet". Because of its high elevation (3,017 m a.s.l.), high accumulation rate (1.4 m w.e. a−1) and thick ice (> 650 m), Eclipse has been the site of several past ice coring campaigns with additional core recovery planned for coming years (McConnell, 2019; Wake et al., 2002; Yalcin et al., 2006). However, recent and continued warming threaten the viability of coring efforts and the preservation of a climate record at Eclipse. Percolating melt can influence*

*paleoproxy records by homogenizing isotope and chemical signals in the snow and firn through which it travels, as well as preclude core recovery with mechanical drills (Moran & Marshall, 2009). Although our results indicate that Eclipse to date remains relatively dry (and could therefore be a valuable ice core site), there is evidence of increasing meltwater inputs and firn modeling shows that firn at the site could become temperate by 2033.*

*We have also moved the hypsometry figure to the appendix per your suggestion.*

**Specific comments**

Line 26: I think Harper et al. (2012) may be a more appropriate reference here.

*Changed reference to Harper et al., (2012)*

Line 29: "as irreducible saturation or a firn aquifer" also as slush fields (e.g., Clerx et al. (2022)) although this isn't really multi-annual storage of meltwater.

*Rephrased to: Surface melt can be retained in the firn pack, either refrozen in ice layers or in liquid form as irreducible water, slush fields, or a firn aquifer*

Line 61: I feel like a bit more description is needed here or needed in this section. Could you define fine grained, coarse grained, textured ice? Perhaps moving sentences on Lines 212 – 215 might be appropriate.

*We have moved the text from lines 212-215 here and added some further language as follows:*

*We describe layers as either "fine-grained" or "coarse-grained". We use the term "fine-grained" to refer to sections of firn composed of <1 mm grains that separate readily when force is applied to the core. In this context, fine-grained firn is firn that shows no visible evidence of melt alteration. We use the term "coarse-grained" to refer to sections of firn composed of 1-2 mm well-sintered ice grains (Fig. 3b). Unlike the large, faceted grains in surface and depth hoar, which reduce layer strength, the rounded, well-sintered grains in our coarse-grained firn result in a strong icy layer requiring a saw to cut through. Despite their sintering, the grains remain readily distinguishable from one another, differentiating these layers from glacier ice. We use the terms "ice lens" and "ice layer" to refer to regions of solid ice that extend partially and fully across our core diameter respectively.*

Line 80: A few things about this line: the parenthetical was a bit confusing to me. Could you just say (i.e., either if the density was greater than 917 kg m-3 or was 300 kg m-3 less than the density of the summer surface). Also, can you add a space between your units here and elsewhere (change kgm-3 to kg m-3). Lastly, why do you plot densities > 917 kg m-3 in Figure 4 if you have removed them as outliers?

*Thanks for the comments. To clarify the text, we rephrased to: We removed outliers from the dataset if the density was greater than $917 \text{ kg m}^{-3}$ or $< 300 \text{ kg m}^{-3}$ below the density of the last summer surface within uncertainty.*

*Spacing in units has been corrected throughout the manuscript. The high density values plotted in Fig. 4 are ones where although the measured value was implausibly high, the uncertainty fell below $917 \text{ kg m}^{-3}$. Values where the full range of uncertainty was above*

*917 kg m⁻³ were the ones that we removed (hence "within uncertainty" included in the text).*

Line 81-82: Is it common to report depths of the top of the segment? Why not report it for the midpoint depth?

> *Density depths are reported in various ways - often to the midpoint or the top and bottom of core segments. We report them here to the top of segments for consistency with our stratigraphy, which we report to segment tops to focus on the transition between melt-altered and unaltered sections of firn, which is more physically interesting than the midpoint depths of sections.*

Line 103-104: The temperature profiles look reasonable, so not a major concern, but even after you let the borehole equilibrate what about any advective heat flow through the top part of the borehole from advective heat transport from wind, etc. while collecting the measurements?

> *Because we were most concerned with temperatures below 10 m depth, boreholes were only a couple inches across, because surface conditions were very similar on both days when temperature measurements were collected (sunny, light breeze), we didn't worry about the effects of advective heat transport for our purposes.*

Line 137: What are typical precipitation patterns like here? Is it reasonable to distribute snow accumulation evenly throughout the year?

> *Snow accumulation in the St. Elias Range is generally dominated by fall and winter precipitation. We have rerun our climate scenario projections with a seasonally variable accumulation rate generated using the mean annual accumulation of 1.4 m w.e. with monthly scalars applied. We calculate these scalars using an in situ snow accumulation record from Divide. We use four years of complete coverage (2004, 2005, 2006, 2008) to compute each month's mean fractional value of annual accumulation. When incorporated into the CFM, we distribute each month's portion of the annual 1.4 m w.e. evenly across the days of the month.*

> *The results for our fifty replicate runs for each climate scenario were consistent with those computed using a mean annual accumulation rate, so we did not rerun all our sensitivity testing (spinup scheme, DDF and surface density).*

Line 150-151: Does the selected pairing of density and DDF also capture the ~1.7 degrees of warming that occurs between 2016 and 2023?

*Not necessarily. We chose to set our 2023 temperature criterion to a range of 2°C because of the >1°C spread of core-bottom temperatures in our field measurements between sites B2 and B5. Given the 2°C criterion range, we excluded the ~1.7°C warming from our model selection criteria.*

Line 158: Where does 0.024 °C yr-1 come from?

*Reference (Williamson et al., 2020) added*

Figure 2: Nice figure. Could you indicate the size of ice lenses by the fractional width that they occupy in the core (similar to Figure 2 in McDowell et. al (2023)? Not a high-priority change.

*We can indicate an estimate of the fractional width, but not with great certainty as we didn't actually measure them (merely sketched in field notes). We didn't do so in the figure because we didn't want to overrepresent our confidence in the fractional width estimate to readers, but can include them with some cautionary language if that would be more helpful than confusing.*

Line 221: Maybe I missed this, but when you say "visibly metamorphized and/or melt altered" are you including ice layers, coarse-grained, and melt-affected sections that you identify in Figure 2?

*Yes! We added the following in parentheses: visibly metamorphized and/or melt altered (ice layers, coarse-grained, and melt-affected sections as shown in Fig. 2)*

Lines 289 – 293: This paragraph is confusing. You state that you "find a significant (p < 0.05) difference" on Line 289. But then on Line 291 you state "we do not find any significant (p < 0.05) difference…". If it is not significant, the p value should be > 0.05 using your confidence level. Also, please check Table B2. It appears that your p values in the table suggest that there are significant differences between median melt season start, end, and length (they are < 0.05)? If that is the case, then these sentences are incorrect, and doesn't that also change your interpretations/conclusions? How do we know if it is melt intensity vs melt duration that is producing temperate firn?

*Ah, we see what's happening with the in-text confusion here - we were trying to indicate that we were using a threshold of p = 0.05 and thinking of the parenthetical as referring specifically to the word "significant" rather than to the whole sentence. That has been changed in the updated text, and all the p values in the tables and elsewhere have been checked and updated as necessary*

*Regardless, we've rerun all the stats after using a seasonally variable accumulation rate, and updated the results, discussion and appendix accordingly. See response to General Comment 1 above.*

Line 317: Should Fig. 11 be referencing Fig. 1 instead?

*Oops, that should be referencing Fig. 12… this has been corrected*

Figure 11: I would consider moving this to the Appendix to make the general text shorter.

*Done*

Line 373: Also probably more appropriate to cite MacFerrin et al. (2019) here.

*Done*

Lines 384-385: Was the 2017 radar profile by McConnell (2019) collected at the same time of year as your radar data? It could have more liquid water just because it was later in the summer?

*Firstly, this should be 2018 (this has been corrected). Methods description has also been updated to include date of acquisition (June 4), which is virtually the same date as our 2023 radar data (June 2-4).*

Lines 453 – 455: Again, please be sure this is correct. It appears that the melt season lengths are significantly different as well.

*See response to General Comment 1 above*

Table B2: See previous comments about the p values. Also, why are the scenarios of warming an order of magnitude lower than what you state in the text on Lines 164 – 165?

*The orders of magnitude were mistyped and have been corrected. Again, sorry! See response to General Comment 1.*

**Reviewer 2**
**Major comments**
**Major Comment 1: suggest to shorten certain sections slightly (e.g., 4.1, 4.2 and 4.4)**

*We've shortened section 4.1 by ~10 lines by eliminating background that can't be directly tied to the observations we're reporting. Edited text below:*

[revised manuscript text omitted]

*We've removed section 4.4 and moved its (greatly shortened) content to an introductory paragraph at the beginning of the Discussion based on the reframing suggested by Reviewer 1:*

> *Results from our firn core stratigraphy, borehole temperatures, and GPR surveys (discussed in greater detail below) suggest that although there is some meltwater movement through the snow and firn at Eclipse, it is not enough to characterize the site as "wet". Because of its high elevation (3,017 m a.s.l.), high accumulation rate (1.4 m w.e. a−1) and thick ice (> 650 m), Eclipse has been the site of several past ice coring campaigns with additional core recovery planned for coming years (McConnell, 2019; Wake et al., 2002; Yalcin et al., 2006). However, recent and continued warming threaten the viability of coring efforts and the preservation of a climate record at Eclipse. Percolating melt can influence paleoproxy records by homogenizing isotope and chemical signals in the snow and firn through which it travels, as well as preclude core recovery with mechanical drills (Moran & Marshall, 2009). Although our results indicate that Eclipse to date remains relatively dry (and could therefore be a valuable ice core site), there is evidence of increasing meltwater inputs and firn modeling shows that firn at the site could become temperate by 2033.*

**Major Comment 2: Improve section about CFM setup**

I'm still a bit confused how you performed the sensitivity runs and the spin-up procedure exactly:

- In Appendix A, you nicely describe the sensitivity of the spin-up to the four different air temperature datasets. However, after only reading Sect. 2.3, I was uncertain if you exclusively use the firn profile generated with the NARR spin-up for all subsequent experiments or not. I would clearly state this in Sect. 2.3 and also briefly mention why you opted for the NARR spin-up.
- Just to be sure – my above assumption is correct, isn't it? You initialise all experiments starting in 2013 with the same firn profile?
- I'm confused which pair of DDF and surface density you use for spinning up the model (à chicken-egg-problem ;-). As I understand from Fig. A2, you derived the optimal pair of DDF and surface density from the period 2013 – 2024. But which pair did you use to derived the initial firn profile for 2013 with the spin-up?

*Yes! You're understanding is correct - we initialize all experiments starting in 2013 with the same profile. And yes there is a bit of a chicken-egg-problem. We try to mitigate this by using the full suite of DDF/surface density pairings to test the spinups and then select the spinup that is generally most conservative overall.*

*Updated text as follows:*

> *We spin the model up from ~1983–2013 (exact spinup time varies slightly among model runs as it is dependent on densification rate and surface melt) using downscaled North American Regional Reanalysis (NARR) air temperatures for Eclipse from 1983 to 2013 (Jarosch et al., 2012). We also test the model sensitivity to three other spinup schemes*

*using different air temperature datasets (Appendix A): 1) elevation-adjusted Divide AWS data from 2013 to 2024 repeated for the duration of the spinup, 2) synthetic climate data selected from a Gaussian distribution of temperatures based on elevation-adjusted 2013–2024 Divide AWS data, and 3) like spinup scheme (2) but with a historical 0.024°Ca⁻¹ rate of temperature change applied for the duration of the spinup such that the mean annual temperature at the start of the main model run is consistent with elevation- adjusted 2013 Divide AWS data. All model spinups are forced with mean annual accumulation rate of 1.4 m w.e.a⁻¹. For our spinup sensitivity tests, we test each spinup scheme with 15 different degree day factor (DDF) values used to estimate surface melt from air temperatures and 20 different surface density values (Table 2). During all sensitivity testing, the model was spun up from 1983 to 2013, and then run through the end of our in situ data in 2024. Across these tests, downscaled NARR temperatures provided the most conservative baseline firn temperatures below 10 m depth.*

*We then use the NARR spinup to explore the model's sensitivity to this suite of DDF and surface density values. Our range of DDF values (2.6–8.2 mm °C⁻¹ d⁻¹) is bounded by the minimum and maximum DDFs derived from 2008–2009 in situ data from two glaciers on the northeast side of the St. Elias Range (MacDougall & Flowers, 2011). Our tested surface density values span the range from 225–520 kg m⁻³, covering the full range of surface snow densities measured at Eclipse (McConnell, 2019), at two sites near the Kaskawulsh/Hubbard Divide (McConnell, 2019; Ochwat et al., 2021), and over three glaciers in the nearby Donjek Range (Pulwicki et al., 2018). Locations and elevations of measured regional surface densities are found in Table 2. We refined the surface density spacing between 400–520 kg m⁻³ since in situ data suggest this is the most reasonable range of surface density estimates for Eclipse. We select a representative pairing (DDF = 6.2, ρ = 450 kg m⁻³) from all the combinations of DDF and surface density values that produce no liquid water down to 14 m depth in the firn in both spring 2016 and spring 2023 (consistent with firn cores and GPR showing no evidence of liquid water at those times), and a firn temperature between –2°C and –4°C at 14 m depth in spring 2023 (consistent with 2023 borehole temperature measurements). We refer to this selected model as our "reference model". Our exploration of model sensitivity to DDF and surface density values is detailed in Appendix A.*

**Major Comment 3: Assumption of time-invariant accumulation rate Line 137: "accumulation rate of 1.4 m w.e. a-1 (McConnell, 2019) distributed evenly throughout the year".**

I wonder if this is a reasonable assumption. To check this, I briefly plotted the monthly precipitation of ERA5 for the closest grid cell: The magnitude of precipitation in autumn/winter is a factor 2-3 larger than in summer, so there exists a distinctive seasonal cycle - of course under the assumption that precipitation from ERA5 (model resolution ~30 km) is representative for the Eclipse side. The annual total agrees well with the measured accumulation rate of 1.4 m w.e. a-1. I think it would be worthwhile to check the sensitivity of the CFM results on seasonally variable precipitation.

*Snow accumulation in the St. Elias Range is generally dominated by fall and winter precipitation, but field data indicates that the seasonal variability is far less than ERA5 suggests (Kindstedt et al. in prep). We have rerun our climate scenario projections with a seasonally variable accumulation rate generated using the mean annual accumulation of 1.4 m w.e. with monthly scalars applied. We calculate these scalars using an in situ snow accumulation record from Divide based on four years of complete coverage (2004, 2005, 2006, 2008) to compute each month's mean fractional contribution to annual accumulation. When incorporated into the CFM, we distribute each month's portion of the annual 1.4 m w.e. evenly across the days of the month.*

*The results for our fifty replicate runs for each climate scenario were consistent with those computed using a mean annual accumulation rate, so we did not rerun all our*

*sensitivity testing (spinup scheme, DDF and surface density). Also see our reply to similar comments from reviewer 1 above.*

**Major Comment 4: Air temperature generator**
I suggest to improve the description of the air temperature generator. Until reading appendix A, I was uncertain how you pool the daily air temperature data (you use daily means, right?) to compute the mean and standard deviation. It seems that you compute these statistics for every day of the year – correct? I would definitely mention this in the main text.

*Thanks for the suggestion. To clarify the text, L165 has been rephrased as follows:*

*We generate synthetic air temperatures for all 2024–2033 climate scenarios by computing the mean and standard deviation of the 2013–2024 Divide AWS daily mean temperature values (computed from hourly measurements) for each day of year. We then assign synthetic daily air temperatures randomly drawn from Gaussian distributions described by these means and standard deviations.*

Furthermore, I wondered how realistic these synthetic air temperature time series actually are. Due to the random selection, there is probably a very high day-to-day fluctuation in air temperature. In reality however, air temperature might sometimes be more constant due to persistent weather patterns.

*It's true that the day-to-day fluctuations are higher in our synthetic timeseries than in our AWS-derived timeseries. However, both of those have **much** higher day-to-day fluctuations than downscaled NARR temperatures that we use for our spinup. We therefore consider the effects of the synthetic timeseries fluctuations minor compared to those of the NARR spinup, with which we were able to produce realistic results at Eclipse (see figure below).*

[Figure]

And finally, I'm astonished by the large spread of surface temperature (ca. 40° C) in Fig. 7b, which is probably related to the air temperature generator. Are the panels to the right showing the temperature for the last day of the simulation (2033-12-31)? Or is temperature averaged over a certain period?

*The panels to the right show all 2033 temperatures, not just from 12/31, so this includes all the nighttime and winter temperatures from 2033, hence the 40°C spread (which also shows up in our surface temperatures derived from AWS air temp measurements 2013-2024). We've added language to the caption to clarify this: "Distributions include all output values over the course of 2033 (Jan 1 through Dec 31)."*

**Minor Comments**
**Content comments**
Line 29: I find the term "irreducible saturation" a bit odd. Maybe better "irreducible water"?

*Changed "irreducible saturation to "irreducible water"*

L30: How exactly do firn aquifers warm the firn?

*As firn aquifers develop, excess cold content in the firn is reduced in the cooling and refreezing of liquid water until no more liquid can refreeze. The aquifer then limits the regeneration of cold content in the surrounding firn.*

L68: I don't understand this sentence: How was the plausible LLS depth (4.0 – 4.5 m) derived from the firn core observations?

*We identified the LSS in our firn cores based on the first appearance of crusty or melt-altered surface. This didn't occur at exactly the same measured depth across our cores, and we needed to correct for core chips and surface lowering, introducing additional uncertainty. 4.0-4.5 m describes the general range that depth observations fell into after corrections were applied. For visual purposes we show the LSS at a depth of 4.25 m. Because we're focusing on the firn below the LSS, standardizing the core depths to the LSS provides a more useful visual than standardizing them to the snow surface.*

L85: I'm unfamiliar with writing error/uncertainty propagation in this way. Do you have a reference for this equation? I'm also confused by the usage of "d" – I guess it is not used for an infinitesimal quantity because later on finite values are assigned to it (e.g., on L91: dL = dD = 0.25). Maybe it's better to replace "d" by the delta symbol?

*We followed Ochwat et al. (2021) to make our data most readily comparable to a similar study in the region. See their Equation 4.*

L99: Why ± 0.2 m? Shouldn't it be ± 0.25 m (in accordance with lines 68/69)?

*As above, lines 68/69 give a general range of our observed LSS depths. The inclusion of this language perhaps made things more, rather than less, confusing so we have rephrased as follows: For visual representation purposes, we show the LSS in all three cores to be located at ~4.25 m depth despite differences in measured depth. Because we're focusing on the firn below the LSS, standardizing the core depths to the LSS provides a more useful visual than standardizing them to the snow surface.*

L103: First I was confused about how the 12 and 1.5 hours fit together. But I assume you let the borehole equilibrate for 12 hours before you start installing/inserting the temperature sensors - right? Maybe you can write this more explicitly.

*Rephrased according to your suggestion: We allowed the borehole to equilibrate for approximately 12 hours after drilling before installing or metering any temperature sensors*

L108: Did you check that 15 s of equilibration time is sufficient (by checking that the measurements during the 30 s are approximately constant)?

*Yes, measured temperature at each depth for 1 minute. Then in analysis, we checked for when temperatures became approximately constant, approaching some limit. This actually should have been written 30 s equilibration time (which we have now updated).*

*We had trimmed 15 s off both ends of the measurement at first in case the sensor was disturbed early toward the end of the 1 minute, but then switched to allowing 30 s of equilibration since we didn't find any evidence of early disruption.*

L113: How is the uncertainty of 0.01° C selected? It seems to be somehow derived from the 15 s equilibrium time…

*We just increased our uncertainty by an order of magnitude in the interest of being conservative when we had less equilibration time.*

L126: Just out of curiosity: how was this semi-automatic picking performed?

*To pick a layer in ImpDAR, the user manually selects points along some identified reflector. The program then interpolates between these points based on a prescribed reflector polarity within some distance from the line segment connecting the two user picks, the distance being frequency-dependent. A full description can be found in Lilien et al., (2020)*

L132: CFM provides multiple densification schemes, why did you choose the one from Kuipers Munneke et al. (2015)?

*There wasn't a particular reason to use that densification equation, but there wasn't a compelling reason to use any other. That equation was calibrated for use in Greenland using firn data from sites spanning a range of climates, and it is reasonable to assume that the climate at Eclipse is similar to some locations in Greenland. Any firn densification model is possibly subject to tuning biases (e.g., Kuipers-Munneke model was tuned using RACMO climate data, so biases in RACMO will affect the model tuning). As such, there is no perfect model for Eclipse in existence, and as such the Kuipers-Munneke model is an appropriate choice. We do not expect that use of a different firn densification equation would change our results and conclusions.*

L132: I was uncertain what you mean by "assigned surface density" until I looked at table 2 and the following text. Maybe you could write here something like: "and a time-invariant surface density derived from a sensitivity test (reference to later text)"

*Rephrased according to your comments as follows: and a time-invariant surface density derived from sensitivity testing (Table 2).*

L134: To which depth did you simulate firn in CFM? Which lower boundary condition for the heat equation was used? Dirichlet or Neuman?

*Rephrased to: We use a parameterization for thermal conductivity from Calonne et al., (2019) and a Neumann boundary condition, simulating down to 50 m depth.*

L148: Although the sensitivity tests are explained in more detail in Appendix A, I would briefly mention some important facts here: over which time was the model run for the sensitivity tests? How was the firn profile initialised for the different sensitivity runs?

*Added the following text:*

*For our spinup sensitivity test, we test each spinup scheme with 15 different degree day factor (DDF) values used to estimate surface melt from air temperatures and 20 different surface density values (Table 2). During all sensitivity testing, the model was spun up from 1983 to 2013, and then run through the end of our in situ data in 2024. Across these tests, downscaled NARR temperatures provided the most conservative baseline firn temperatures below 10 m depth. We then use the NARR spinup to explore the model's sensitivity to the full suite of DDF and surface density values.*

L153: How is spin-up time defined? The time required to refresh the entire simulated firn column?

*Exactly, the time required to refresh the entire firn column, which happened to be ~30 years.*

L172: The division in section 2.4 and 2.5 is not entirely clear to me – maybe one could list all reference data in one section (e.g., as bullet points)

*The goal here was to make it clear which data pertained to spatial comparisons (i.e. reference data not from Eclipse) and which data pertained to temporal comparisons (i.e. from earlier studies at Eclipse). We think there is value in separating these out so as to not get our comparisons mixed up, but we're not doing a great job of that if the division is causing more rather than less confusion. We've retitled section 2.5 "Firn changes at Eclipse over time" to improve this.*

L262: How do you infer an ice thickness of only ~150 m from Fig. 6c?

*Good catch! This was a typo. Changed to ~400 m*

L266: Explain what you mean by "reference model"

*Reference model is defined in Appendix A as the model we selected from our sensitivity test results for all subsequent model runs. We have added this definition to the main manuscript (section 2.3).*

L277: Are the simulations shown in Fig. 9 just random examples from the 50 members? Or were they specifically selected?

*These are random examples. Text and caption edited as follows to capture this:*

*In-text:*
*The evolution of density, liquid water content and temperature over the full firn column is shown under each modeled climate scenario in Figure 9 for a model run randomly selected from the fifty replicates.*

*Caption:*
*Example timeseries of density, liquid water content and firn temperature from 2013–2033 under six different climate scenarios: 0.1°C cooling by 2033 (a-b), continuation of current climate through 2033 (c-d), 0.1°C warming by 2033 (e-f), 0.2°C warming by 2033 (g-h), 0.5°C warming by 2033 (i-j), and 1°C warming by 2033 (k-l). The righthand panels show both density and LWC, with density shown by the colorbar and white areas indicating the presence of liquid water. All climate scenarios are prescribed using surface temperature values drawn from a Gaussian distribution based on elevation-corrected 2013–2024 AWS data from Divide with the appropriate level of warming or cooling applied.*

L280: I don't understand this sentence, could you rephrase it?

*This sentence has been removed after rerunning and analyzing our results. See response to Reviewer 1 General Comment 1 for an account of these revisions*

L309: "and less than" → "and more than"?

*Yes, good catch! Changed as suggested.*

L327: What do you mean by "peaks and cyclic variations"?

*"Peaks" refers to specific high values, while "cyclic variations" refers to the pattern of cycling from high to low then back to high density. Rephrased to: however, both the location of peaks and transitions from periods of high to low measured density are consistent with our stratigraphic observations.*

L389: I do not fully understand this sentence, could you rephrase it?

*We deleted this sentence in the reworking of section 4.2 (text included in response to Major Comment 1)*

L418: How do you know that mean annual temperature at the two sites is virtually the same?

*The sites are only 1.5 km apart, nearly identical in elevation, and neither is substantially more exposed than the other. We also edited this language slightly (text included in response to Major Comment 1)*

L421: I'm confused: B5 receives more solar radiation than B2 but shows nonetheless fewer melt features?

> *The idea here is that B5 receives more solar radiation, but preferentially in the winter because of its aspect. The additional radiation may be enough to warm the surface some, but not enough to produce sufficient melt to be recorded as melt features. We suggest this as a possible but not necessarily probable explanation for the "fewer melt features but warmer firn" phenomenon.*

L444: "is < 2° C from supporting liquid water at depth" → could you rephrase that?

> *Rephrased to: Despite remaining dry to date, borehole temperatures indicate that at least part of Eclipse Icefield is < 2°C from the melting point at depth.*

L453: Now I'm confused: What's the difference between number of PDD and number of melt events? It's probably helpful if you introduce and explain these metrics somewhere.

> *PDDs quantify the amount of warmth above 0°C, while melt events are are periods of time where the snow surface is continuously melting. One melt event can therefore span several PDDs. We define melt events on L284 ("We define melt events as any period over which the snow surface is continuously melting without refreeze."). PDDs are a standard metric, so we don't give an extended description for brevity's sake.*

L469: "Additionally, the loss of firn pore space…" → difficult to understand, could you rephrase this?

> *Rephrased as follows:*

> *Additionally, firn loses pore space in response to warming more readily than it gains pore space in response to cooling; observed densification of the firn to date therefore has long-term consequences for runoff buffering (Thompson-Munson et al., 2024).*

L534: I'm not familiar with the Kruskal-Wallis test. Could you briefly explain what "H" represents?

> *Added the following text: The Kruskal-Wallis test is the non-parametric equivalent of ANOVA; it is used to test whether two or more samples originate from the same distribution. A higher H value reflects a larger difference between the medians of the samples in question, meaning it is more likely they are from different distributions.*

Figure 6: Is the abbreviation "TWTT" explained somewhere?

> *Added to the figure caption: Two-way travel time (TWTT, lefthand axis) is used to calculate depth (righthand axis).*

Figure 9: Colorbar for temperature not very intuitive (transition from yellow to red/violet is normally interpreted as warming…)

*The colorbar has been changed so that the transition from yellow to red represents warming.*

Figure 11: Is the Colorbar of (a) identical to the one used for (b)? Because according to this colorbar, the north-eastern region seems very low elevated (~0 m) but it is higher in reality I guess…

*The colorbars are identical. Elevations are only shown for glacierized regions. Areas of pure white are outside of our region of analysis (i.e. they are not glacier covered). The caption has been updated to include this.*

Table 1: What limited the (different) bottom depths of the three recovered firn cores?

*Core B501 was drilled to the extent of our drill cable. Drilling of B502 was limited by the time we had that night after drilling B501. Core B201 was limited by mechanical difficulties, where the drill seemed to stop operating at full power and couldn't bite through the ice layer it had encountered. We haven't added this information to the manuscript as we don't believe that it's relevant for the data that we're presenting.*

Figure A1: Why is there a range of simulation for the 4 experiments? Did you vary something else besides the four driving air temperature datasets?

*We varied degree day factor (DDF) and surface density values. Phrasing from earlier response above:*

*We test each spinup scheme with 15 different degree day factor (DDF) values used to estimate surface melt from air temperatures and 20 different surface density values (Table 2). During all sensitivity testing, the model was spun up from 1983 to 2013, and then run through the end of our in situ data in 2024. Across these tests, downscaled NARR temperatures provided the most conservative baseline firn temperatures below 10 m depth. We then use the NARR spinup to explore the model's sensitivity to the full suite of DDF and surface density values.*

**Typos, phrasing and stylistic comments**
L80: space missing between "kg" and "m-3" (twice)

*Corrected here and throughout manuscript*

L146: "We tested a higher concentration of surface density values…" → "We refined the surface density spacing between 400 – 520 kg m-3 since…"

*Done*

L146: space missing between "kg" and "m-3"

*Corrected here and throughout manuscript*

L151: I would remove "to predict the evolution of the firn pack from 2024 – 2033"

*Done*

L158: space missing between "C" and "a-1"

*Corrected here and throughout manuscript*

L174: no space between "m" and "a.s.l." (same on line 176)

*Corrected here and throughout manuscript*

L186: "between the our" → "between our"

*Done*

Figure A4: "…between 1979 and 2016 applied" → rephrase

*Rephrased as follows:*

*Panel (b) shows model runs spun up with temperature values randomly selected from a Gaussian distribution based on elevation-corrected Divide AWS data. A historical warming rate of 0.024°C $a^{-1}$ between 1979 and 2016 was applied to these data (Williamson et al., 2020).*

Table B2: Decimal place errors in first column. E.g., "+0.05° C" → "+0.5° C"

*Corrected*

---

## Referee Report (RR1)

I would like to thank the authors for the additional effort they put in the revision of the manuscript. Please find below some additional final remarks (line numbers refer to the revised manuscript with marked changes):

**Day-to-day fluctuation in air temperature**
Thanks for the additional graph in the replies comparing NARR, AWS and synthetic data. I'm a bit puzzled now about the distinctively different (and lower) fluctuation in the NARR data. Did you use monthly averaged air temperature data from NARR? If so, I would mention this somewhere…

**L127:** "we increase our uncertainty…" → still difficult to understand – I would rather remove this part. The following new sentence is also a bit hard to follow: Why is the borehole diameter (better use SI units to specify) related to the depth of measured temperatures?
**L157:** Maybe better: "and a Neuman boundary condition for the heat equation at the bottom of the 50 m deep firn column."
**L174:** Very nice that you now consider seasonal variable snow accumulation. Could you show the monthly scalars you computed (respectively the seasonal variation in snow accumulation) in a plot (could be moved to the appendix)?
**L490:** Sorry, I'm still confused by your reasoning. I think I can (partially) follow why aspect and solar radiation could contribute to a slightly warmer temperature – but why does this configuration cause less melt in summer (B5 should then still receive more radiation – right)? If in doubt, I would rather remove this part because I think the meltwater percolation hypothesis is sufficient…
**L542:** Forgot to include rephrased sentence: *Additionally, firn loses pore space in response to warming more readily than it gains pore space in response to cooling; observed densification of the firn to date therefore has long-term consequences for runoff buffering (Thompson-Munson et al., 2024).*
Overall, I'm still a bit puzzled by this statement: I guess the rate of firn pore space increase during cooling is, after reaching sufficiently cold temperatures, mainly a function of snow accumulation, which is not mention here…
**L610:** Spell out abbreviation "ANOVA"

**Fig. A4:** Forgot to include rephrased sentence: *Panel (b) shows model runs spun up with temperature values randomly selected from a Gaussian distribution based on elevation-corrected Divide AWS data. A historical warming rate of 0.024°C a$^{-1}$ between 1979 and 2016 was applied to these data (Williamson et al., 2020).*

**Typos, phrasing and stylistic comments**
**L14:** "would represent likely indicate" → would likely indicate"
**L76:** "we're" → "we are"
**L88:** "We removed outliers…" → this sentence reads odd and should be rephrased
**L154:** Better (?): "We do not expect that using a different firn"
**L201:** "We" should remain in the sentence.
**L455:** "can be observed Eclipse at the kilometre scale…" → fix

---

## Referee Report (RR2)

**Re-review of Kindstedt et al.:** *Ongoing firn warming at Eclipse Icefield, Yukon, indicates potential widespread meltwater percolation and retention in firn pack across the St. Elias Range*
* * *
I would like to thank the authors for engaging seriously with the reviews from both me and the other reviewer. I think the paper has been improved and would make a good addition to The Cryosphere. Many of my comments are stylistic or technical; however, I have three main comments that I think should be considered before the manuscript is ready for publication.

**Main Comments:**

**[1]** I would encourage the authors to define "extreme melt events" or "extreme individual melt events". It appears the author's definition of "extreme" is "high-intensity". What is the intensity threshold that makes it extreme? It may be better to just say "intense melt events" rather than extreme, as in Greenland, extreme melt events commonly refer to the melt extent.

**[2]** In the abstract the authors state, *"…the development of year-round deep temperate firn at Eclipse Icefield is promoted by an increase in extreme individual melt events, rather than a greater number of small melt events or a prolonged melt season"* (**L6-8**). I still am a bit unclear on how the authors arrive at that conclusion. Across most years, the median PDDs, number of melt events, and melt event magnitude are all significantly greater in model runs that produce temperate firn (Table B2), so why zero-in on the extreme individual events? How do the authors know that is driving firn warming more than the number of melt events?

I think Figure 10 could be improved by making this a 3-panel plot (although I'm happy to consider other changes to the plot that the authors see fit). Similarly to how PDDs are displayed currently, additional panels could show the distribution of number of melt days between models that produce temperate firn and those that don't, as well as the distribution of melt intensity per melt event between models that produce temperate firn and those that don't. Since these are the three significant drivers producing temperate firn, it would be nice to see them displayed in a figure, and it may allow the authors to make their point clearer about why more intense melt is the main driver.

As it stands, the authors state their conclusions slightly differently throughout the text. Just to highlight, on **L457-459** in the Discussion, the authors state: *"Model results for Eclipse show the development of year-round temperate firn at 15 m depth associated with an increase in total PDDs throughout the melt season, as well as with a greater number and more extreme melt events, rather than an earlier or prolonged melt season"*. That is slightly different from the abstract, though better supported by the statistics. Lastly, in the Conclusion on **L489-490**, the authors state *"Development of year-round temperate firn at Eclipse is associated with an increase in total PDDs throughout the melt season and more extreme individual melt events rather than a greater number of melt events or prolonged melt season."* This appears to be more similar to what is stated in the introduction. I would try to be as consistent as possible.

**[3]** I appreciate the revision of the tables in Appendix B. The editor and authors may feel this is unnecessary, but I am partial to reporting the true p-values to the readers. P-values that are significant

below the author's confidence threshold could be highlighted or in bold text. Also, it may be nice to report the statistics being evaluated (e.g. difference in median PDDs, difference in median number of melt events, and difference in median melt event magnitude). It may make the authors conclusions clearer and less "hidden" behind the statistics. I would also consider moving these to the main text since they directly support the authors' conclusions. I think there are some figures/tables that could be moved to the appendix to make space (e.g., Figure 3, Table 1, Table 2).

**Minor Comments**

- **L14:** Remove the word "represent" where it says *"... would represent likely indicate…"*
- **L30:** *"Firn aquifers account for much of observed firn water storage"...* Much seems a little vague here. Maybe the authors could provide an estimate? An approximate water storage volume or areal extent of firn aquifers compared to a melt area extent? Not a big deal but could strengthen the sentence.
- Upon rereading the manuscript, I'm wondering if in the paragraph from **L176-188** along with **Table 2**, it may be appropriate to move to the appendix since it is more detailed sensitivity tests that, while interesting and useful, could be available for interested readers at the end, which would shorten some of the main text. I think it also will keep more focus on running CFM under a suite of climate scenarios as the authors describe in the following paragraph.
- **Section 3.1.** I would recommend including a sentence in the beginning paragraph to highlight what the authors would like for readers to take away from this. It gets quite dense when describing the detailed stratigraphy in each paragraph. Maybe just highlight the key point for readers to have something to hold onto… maybe indicating that these measurements are important to demonstrate the substantial variability between cores, even spaced less than 1 m apart?
- **Section 3.2.** A similar suggestion as above. The authors may even be able to start with the sentences: *"In general, density increases with depth throughout the core. However, cyclic variations can be seen, which are likely seasonal, particularly in the top 10 m. Individual ice layers can also be identified by peaks in density"* and reference Figure 4. I think it is helpful to know what the authors want the reader to see in the figure when referencing it. As it stands, the initial sentence to start 3.2 does not really provide any information for this section.
- **Line 298:** Where the authors say "at least some" could they just say how many runs?
- **Line 327:** Show the same results as in 2016? Maybe clarify this in the sentence.

---

## Author Response (AR2)

**Reviewer 1**

I would like to thank the authors for the additional effort they put in the revision of the manuscript. Please find below some additional final remarks (line numbers refer to the revised manuscript with marked changes):

**Day-to-day fluctuation in air temperature**
Thanks for the additional graph in the replies comparing NARR, AWS and synthetic data. I'm a bit puzzled now about the distinctively different (and lower) fluctuation in the NARR data. Did you use monthly averaged air temperature data from NARR? If so, I would mention this somewhere…

*Text edited to:*

> *We spin the model up from ~1983–2013 (exact spinup time varies slightly among model runs as it is the time required to refresh the entire firn column and therefore dependent on densification rate and surface melt) using monthly averaged downscaled North American Regional Reanalysis (NARR) air temperatures for Eclipse from 1983 to 2013 Jarosch et al. 2012).*

L127: "we increase our uncertainty…" → still difficult to understand – I would rather remove this part. The following new sentence is also a bit hard to follow: Why is the borehole diameter (better use SI units to specify) related to the depth of measured temperatures?

*The borehole diameter is not related to the depth of measured temperatures. Rather, both of those elements are related to our choice to ignore advective heat transport. Thanks for pointing out the confusion here; we've rearranged the text as follows to clarify:*

> *We assign an uncertainty of 0.01°C to our temperature profiles at 20 cm increments when a 30 s equilibrium time was used. We ignore the effects of advective heat transport for three reasons: we were most concerned with temperatures below 10 m depth, boreholes were only ~5 cm across, and surface conditions were very similar on both days when temperature measurements were collected (sunny, light breeze).*

L157: Maybe better: "and a Neuman boundary condition for the heat equation at the bottom of the 50 m deep firn column."

*Changed as suggested*

L174: Very nice that you now consider seasonal variable snow accumulation. Could you show the monthly scalars you computed (respectively the seasonal variation in snow accumulation) in a plot (could be moved to the appendix)?

*We have added the following figure:*

[Figure]

**Figure B5.** Monthly accumulation scalars calculated using an in situ snow accumulation record from Divide. We use four years of complete coverage (2004, 2005, 2006, 2008) to compute each month's mean fractional contribution to annual accumulation. We compute the fractional contribution to total positive surface change (ignoring ablation) to represent snowfall rather than net accumulation.

L490: Sorry, I'm still confused by your reasoning. I think I can (partially) follow why aspect and solar radiation could contribute to a slightly warmer temperature – but why does this configuration cause less melt in summer (B5 should then still receive more radiation – right)? If in doubt, I would rather remove this part because I think the meltwater percolation hypothesis is sufficient…

*We included both hypotheses because each has a notable flaw. As you point out, it is unlikely for the higher solar radiation at B5 to be entirely a wintertime phenomenon, but Eclipse is high enough latitude that the sun position in the sky is considerably further south in the winter than summer, so this may play some role. The main flaw in the meltwater percolation hypothesis is the probably limited lateral transport of liquid meltwater in cold firn before it refreezes. We've modified the text as follows to more explicitly address the flaw in the first hypothesis that you point out. We do think both hypotheses are valuable to keep in the text because the meltwater percolation hypothesis doesn't seem sufficient to us.*

> *One explanation for observed differences between sites B2 and B5 is that B5 on average receives slightly more solar radiation due to its southeast aspect, especially in the winter. Eclipse is high enough latitude that the sun position in the sky is considerably further south in the winter than summer (approximately a 30° difference in azimuth between noon on 21 June and noon on 21 December). This being a predominantly winter phenomenon could explain why B5 has fewer melt features despite being slightly warmer; however, we consider it unlikely that the wintertime southerly migration of sun position would outweigh the influence of summertime melt production at the two sites. Moreover, in areas with surface melt and percolation into the subsurface, the role of conduction in downward heat transport is comparatively minor relative to that of latent heat associated with the refreezing of meltwater (Cuffey and Paterson, 2010).*

L542: Forgot to include rephrased sentence: Additionally, firn loses pore space in response to warming more readily than it gains pore space in response to cooling; observed densification of the firn to date therefore has long-term consequences for runoff buffering (Thompson-Munson et al., 2024). Overall, I'm still a bit puzzled by this statement: I guess the rate of firn pore space increase during cooling is, after reaching sufficiently cold temperatures, mainly a function of snow accumulation, which is not mention here…

*The rephrased sentence has now been included! Yes, we would agree that new snow accumulation would likely increase pore space if it can be retained without melting, "resetting" some of the firn column. However, the paper we reference here presents idealized warming and cooling experiments on Greenland's firn, focusing on the effects of air temperature on compaction and melt. We only mention it in passing to point out that the effects of firn densification in response to atmospheric warming probably aren't easily reversible and should be taken seriously.*

L610: Spell out abbreviation "ANOVA"

*Done*

Fig. A4: Forgot to include rephrased sentence: Panel (b) shows model runs spun up with temperature values randomly selected from a Gaussian distribution based on elevation-corrected Divide AWS data. A historical warming rate of 0.024°C a$^{-1}$ between 1979 and 2016 was applied to these data (Williamson et al., 2020).

*The rephrased sentence has now been included*

**Typos, phrasing and stylistic comments**
L14: "would represent likely indicate" à would likely indicate"

*Done*

L76: "we're" → "we are"

*Done*

L88: "We removed outliers…" → this sentence reads odd and should be rephrased

*Rephrased to:*

*We removed outliers for all depths below the last summer surface if their density was > 917 kg m$^{-3}$ or < 300 kg m$^{-3}$ within uncertainty.*

L154: Better (?): "We do not expect that using a different firn"

*Changed as suggested*

L201: "We" should remain in the sentence.

*Double checked that "we" is in the sentence*

L455: "can be observed Eclipse at the kilometre scale…" → fix"

*Changed to: "can be observed at Eclipse…"*

**Reviewer 2**

I would like to thank the authors for engaging seriously with the reviews from both me and the other reviewer. I think the paper has been improved and would make a good addition to The Cryosphere. Many of my comments are stylistic or technical; however, I have three main comments that I think should be considered before the manuscript is ready for publication.

**Main Comments:**
[1] I would encourage the authors to define "extreme melt events" or "extreme individual melt events". It appears the author's definition of "extreme" is "high-intensity". What is the intensity threshold that makes it extreme? It may be better to just say "intense melt events" rather than extreme, as in Greenland, extreme melt events commonly refer to the melt extent.

*We have changed our phrasing throughout to avoid confusion with extreme melt events in Greenland. You are correct in that we are talking about high-intensity melt events. Specifically, we're talking about an increase in the average individual melt event magnitude (mm melt produced). We have changed our language as follows to clarify this:*

> *We suggest that more and higher-magnitude melt events during the height of summer combined with warmer wintertime temperatures promote the development of year-round temperate firn in the St. Elias. Model results for Eclipse show the development of year-round temperate firn at 15 m depth associated with an increase in total PDDs throughout the melt season, as well as with a greater number of individual melt events, higher average melt event magnitude (mm melt produced), and warmer winter temperatures, rather than an earlier or prolonged melt season (Fig. 8; Tables 4, 5). In Greenland, "extreme melt events" have been related to firn's multi-year response to surface melt via the formation of thick ice slabs and ice layer complexes, which cause a near-surface barrier to downward percolation (Culberg et al., 2021). In the St. Elias, however, an increase in the number of melt events and in average individual melt event magnitude are more likely to result in sustained heat transport to depth because of the insulating effect of the region's high annual accumulation (1.4 m w.e. $a^{-1}$ at Eclipse) relative to accumulation rates in Greenland (0.3 – 1.2 m w.e. $a^{-1}$; Hawley et al., 2020; Montgomery et al., 2020; Burgess et al., 2010).*

[2] In the abstract the authors state, "…the development of year-round deep temperate firn at Eclipse Icefield is promoted by an increase in extreme individual melt events, rather than a greater number of small melt events or a prolonged melt season" (L6-8). I still am a bit unclear on how the authors arrive at that conclusion. Across most years, the median PDDs, number of melt events, and melt event magnitude are all significantly greater in model runs that produce temperate firn (Table B2), so why zero-in on the extreme individual events? How do the authors know that is driving firn warming more than the number of melt events?

*Good catch! This is a holdover error that should have been corrected with the revision of the tables in Appendix B (now in the main text). The abstract text has been rephrased as follows:*

> *In particular, the development of year-round deep temperate firn at Eclipse Icefield is promoted by an increase in the number of individual melt events and in average melt event magnitude combined with warmer wintertime temperatures, rather than an earlier or prolonged melt season.*

I think Figure 10 could be improved by making this a 3-panel plot (although I'm happy to consider other changes to the plot that the authors see fit). Similarly to how PDDs are displayed currently, additional panels could show the distribution of number of melt days between models that produce temperate firn and those that don't, as well as the distribution of melt intensity per melt event between models that produce temperate firn and those that don't. Since these are the three significant drivers producing temperate firn, it would be nice to see them displayed in a figure, and it may allow the authors to make their point clearer about why more intense melt is the main driver.

*Figure 10 has been converted into the following 4-panel plot (we included winter temps in addition to you three suggested panels):*

[Figure]

**Figure 8.** Melt season positive degree days (PDDs, a), number of melt events (b), melt event magnitude (c), and wintertime temperatures (d) each year for model runs that produce year-round temperate firn at 15 m depth by 2033 (blue) and those that do not (orange). We define the melt season here as May-September, and wintertime as October-March. Outliers are excluded for clarity.

As it stands, the authors state their conclusions slightly differently throughout the text. Just to highlight, on L457-459 in the Discussion, the authors state: "Model results for Eclipse show the development of year-round temperate firn at 15 m depth associated with an increase in total PDDs throughout the melt season, as well as with a greater number and more extreme melt events, rather than an earlier or prolonged melt season". That is slightly different from the abstract, though better supported by the statistics. Lastly, in the Conclusion on L489-490, the authors state "Development of year-round temperate firn at Eclipse is associated with an increase in total PDDs throughout the melt season and more extreme individual melt events rather than a greater number of melt events or prolonged melt season." This appears to be more similar to what is stated in the introduction. I would try to be as consistent as possible.

*The conclusion text has been edited to be consistent with the updated abstract and discussion as follows:*

> *Development of year-round temperate firn at Eclipse is associated with more total PDDs throughout the melt season, more individual melt events, and a higher average melt*

*event magnitude combined with warmer wintertime temperatures, rather than with an earlier or prolonged melt season.*

[3] I appreciate the revision of the tables in Appendix B. The editor and authors may feel this is unnecessary, but I am partial to reporting the true p-values to the readers. P-values that are significant below the author's confidence threshold could be highlighted or in bold text. Also, it may be nice to report the statistics being evaluated (e.g. difference in median PDDs, difference in median number of melt events, and difference in median melt event magnitude). It may make the authors conclusions clearer and less "hidden" behind the statistics. I would also consider moving these to the main text since they directly support the authors' conclusions. I think there are some figures/tables that could be moved to the appendix to make space (e.g., Figure 3, Table 1, Table 2).

*The tables have been changed to those shown below, and moved to the main text.*

**Table 3.** Difference in mean winter (October through March) temperatures (T, independent T-test) between model runs that produce year-round temperate firn by 2033 and those that do not. Significant results (at the 95% confidence level) are bolded.

| Year | Temperate mean (°C) | Polythermal mean (°C) | T | p |
|------|--------------------|-----------------------|-------|-------------------------|
| 2024 | -16.63 | -16.66 | 0.77 | 0.44 |
| 2025 | -16.46 | -16.53 | 1.49 | 0.13 |
| 2026 | **-16.40** | **-15.54** | **2.84** | $\mathbf{4.5 \times 10^{-3}}$ |
| 2027 | **-16.35** | **-16.51** | **3.34** | $\mathbf{8.3 \times 10^{-4}}$ |
| 2028 | **-16.27** | **-16.55** | **5.89** | $\mathbf{4.0 \times 10^{-9}}$ |
| 2029 | **-16.20** | **-16.56** | **7.48** | $\mathbf{7.6 \times 10^{-14}}$ |
| 2030 | **-16.08** | **-16.50** | **8.87** | $\mathbf{7.6 \times 10^{-19}}$ |
| 2031 | **-16.07** | **-16.49** | **8.78** | $\mathbf{1.6 \times 10^{-18}}$ |
| 2032 | **-16.01** | **-16.50** | **10.23** | $\mathbf{1.5 \times 10^{-24}}$ |
| 2033 | **-15.88** | **-16.44** | **11.94** | $\mathbf{8.0 \times 10^{-33}}$ |

**Table 4.** Difference in median number and magnitude of melt events (W, Wilcoxon Rank Sum tests) between model runs that produce year-round temperate firn by 2033 and those that do not. Significant results (at the 95% confidence level) are bolded.

| Year | Median PDDs W | Median PDDs p | Number of melt events W | Number of melt events p | Melt event magnitude W | Melt event magnitude p |
|------|------|------|------|------|------|------|
| 2024 | -0.13 | 0.90 | 0.02 | 0.98 | 0.27 | 0.78 |
| 2025 | -0.10 | 0.92 | 0.81 | 0.442 | $4.2 \times 10^{-3}$ | 1.0 |
| 2026 | **4.68** | $\mathbf{2.9 \times 10^{-6}}$ | 1.46 | 0.14 | **2.74** | $\mathbf{6.1 \times 10^{-3}}$ |
| 2027 | **4.73** | $\mathbf{2.2 \times 10^{-6}}$ | **3.33** | $\mathbf{8.6 \times 10^{-4}}$ | **3.03** | $\mathbf{2.4 \times 10^{-3}}$ |
| 2028 | **4.72** | $\mathbf{2.3 \times 10^{-6}}$ | 0.93 | 0.35 | 1.17 | 0.24 |
| 2029 | **6.01** | $\mathbf{1.7 \times 10^{-9}}$ | **3.15** | $\mathbf{1.6 \times 10^{-3}}$ | **3.06** | $\mathbf{2.2 \times 10^{-3}}$ |
| 2030 | **7.63** | $\mathbf{2.3 \times 10^{-14}}$ | **2.46** | **0.01** | **3.81** | $\mathbf{1.4 \times 10^{-4}}$ |
| 2031 | **8.87** | $\mathbf{7.6 \times 10^{-19}}$ | **4.87** | $\mathbf{1.1 \times 10^{-6}}$ | **5.47** | $\mathbf{4.6 \times 10^{-8}}$ |
| 2032 | **9.16** | $\mathbf{5.3 \times 10^{-20}}$ | **4.21** | $\mathbf{2.6 \times 10^{-5}}$ | **5.45** | $\mathbf{5.1 \times 10^{-8}}$ |
| 2033 | **8.26** | $\mathbf{1.4 \times 10^{-16}}$ | **4.37** | $\mathbf{1.2 \times 10^{-5}}$ | **3.21** | $\mathbf{1.3 \times 10^{-3}}$ |

**Table 5.** Difference in median melt season timing and duration (W, Wilcoxon Rank Sum tests) between model runs that produce year-round temperate firn by 2033 and those that do not. No years showed significant differences between the medians at the 95% confidence level.

| Year | Melt season start W | Melt season start p | Melt season end W | Melt season end p | Melt season length W | Melt season length p |
|------|------|------|------|------|------|------|
| 2024 | -0.18 | 0.86 | 0.87 | 0.38 | 0.87 | 0.39 |
| 2025 | 0.24 | 0.81 | 0.89 | 0.37 | -0.15 | 0.88 |
| 2026 | 1.27 | 0.21 | -1.65 | 0.99 | -1.66 | 0.10 |
| 2027 | 1.36 | 0.17 | -1.90 | 0.06 | -1.72 | 0.09 |
| 2028 | 0.90 | 0.37 | 0.34 | 0.74 | -0.44 | 0.66 |
| 2029 | -0.02 | 0.98 | -0.67 | 0.50 | -1.32 | 0.19 |
| 2030 | -0.21 | 0.83 | 1.55 p | 0.12 | 1.50 | 0.13 |
| 2031 | 0.41 | 0.68 | -1.15 | 0.25 | -1.55 | 0.12 |
| 2032 | 1.20 | 0.23 | 1.06 | 0.29 | -0.36 | 0.72 |
| 2033 | -0.74 | 0.46 | 1.08 | 0.28 | 1.37 | 0.17 |

**Minor Comments**

L14: Remove the word "represent" where it says "… would represent likely indicate…"

*Done*

L30: "Firn aquifers account for much of observed firn water storage"… Much seems a little vague here. Maybe the authors could provide an estimate? An approximate water storage

volume or areal extent of firn aquifers compared to a melt area extent? Not a big deal but could strengthen the sentence.

*We've edited the text as follows to provide some quantitative grounding:*

> *Firn aquifers can store large amounts of liquid water, and can retain water for several years, both delaying runoff and warming the firn (Ochwat et al., 2021; Miège et al., 2016; Jansson et al., 2003; Schneider, 1999; Fountain, 1989). For example, firn aquifers across Greenland have been estimated to store 140 ± 20 Gt of liquid water, buffering 0.4 mm of sea level rise (Koenig et al., 2014).*

Upon rereading the manuscript, I'm wondering if in the paragraph from L176-188 along with Table 2, it may be appropriate to move to the appendix since it is more detailed sensitivity tests that, while interesting and useful, could be available for interested readers at the end, which would shorten some of the main text. I think it also will keep more focus on running CFM under a suite of climate scenarios as the authors describe in the following paragraph.

*Done*

Section 3.1. I would recommend including a sentence in the beginning paragraph to highlight what the authors would like for readers to take away from this. It gets quite dense when describing the detailed stratigraphy in each paragraph. Maybe just highlight the key point for readers to have something to hold onto… maybe indicating that these measurements are important to demonstrate the substantial variability between cores, even spaced less than 1 m apart?

*We have changed the first sentence to the following:*

> *The stratigraphy of all three 2023 cores shows ice layers, ice lenses, and melt-affected firn throughout the core; however, variability among individual melt features is high among all three cores despite the proximity (<1 m) of cores B501 and B502 230 (Fig. 2).*

Section 3.2. A similar suggestion as above. The authors may even be able to start with the sentences: "In general, density increases with depth throughout the core. However, cyclic variations can be seen, which are likely seasonal, particularly in the top 10 m. Individual ice layers can also be identified by peaks in density" and reference Figure 4. I think it is helpful to know what the authors want the reader to see in the figure when referencing it. As it stands, the initial sentence to start 3.2 does not really provide any information for this section.

*We've rearranged the text according to your suggestion, beginning the paragraph with:*

> *In general, density increases with depth throughout the core. However, cyclic variations can be seen, which are likely seasonal, particularly in the top 10 m. Individual ice layers can also be identified by peaks in density (Fig. 3).*

Line 298: Where the authors say "at least some" could they just say how many runs?

*We could, but it gets a bit unwieldy to report the number of runs for every depth just to make the point that the number is non-zero. We chose not to report for each depth here, so that when we do report specific numbers for each climate scenario later on, it helps emphasize that's the bigger point.*

Line 327: Show the same results as in 2016? Maybe clarify this in the sentence.

*Line 327 states "Conditions associated with the development of year-round temperate firn at 15 m depth include lower mean winter temperature, higher total melt season positive degree days (PDDs), more individual melt events, and higher average melt event intensity."*

*We're unsure where the reference to 2016 is coming from. Perhaps the line number is a typo?*

**Technical corrections**

Please ensure that the colour schemes used in your maps and charts allow readers with colour vision deficiencies to correctly interpret your findings. Please check your figures using the Coblis – Color Blindness Simulator (https://www.color-blindness.com/coblis-color-blindness-simulator/) and revise the colour schemes accordingly. --> Figs. 7, 8, A2, and A3

*We have used the color blindness simulator to check all of our figures, including the ones mentioned here. Because there are fifty individual lines in these panels, it is impossible to easily distinguish all of them, even without any form of color blindness. However, the point here is to show the distribution and general range of variability among model runs, and these elements can be discerned.*

---

## Author Response (AR3)

In general, your revisions appear adequate except in one instance.

In the tracked changes version, Line 434, you write:

"Eclipse is high enough latitude that the sun position in the sky is considerably further south in the winter than summer (approximately a 30° difference in azimuth between noon on 21 June and noon on 21 December)."

I do not understand this. Azimuth is a horizontal angle and is always the same at noon. I must have missed something. Can you please elaborate and possibly correct?

*Thank you for correcting this. We have removed the azimuth statement and changed the sentence to focus on sunrise:*

*One explanation for observed differences between sites B2 and B5 is that B5 on average receives slightly more solar radiation due to its southeast aspect, especially in the winter. Eclipse is high enough latitude that the sun rises considerably further south in the winter than summer.*